# Abundance of non-conservative microplastics in the upper ocean from 1957 to 2066

Atsuhiko Isobe[1], Shinsuke Iwasaki[2], Keiichi Uchida[3] & Tadashi Tokai[3]

Laboratory-based studies have suggested that marine organisms can be harmed by ingesting microplastics. However, unless the current and future microplastic abundance in the ocean environment is quantified, these experimental studies could be criticized for using an unrealistic density or sparsity of microplastics. Here we show the secular variations of pelagic microplastic abundance in the Pacific Ocean from 1957 to 2066, based on a combination of numerical modeling and transoceanic surveys conducted meridionally from Antarctica to Japan. Marine plastic pollution is an ongoing concern especially in the North Pacific, and pelagic microplastics are regarded as non-conservative matter due to the removal processes that operate in the upper ocean. The results of our numerical model incorporating removal processes on a 3-year timescale suggested that the weight concentrations of pelagic microplastics around the subtropical convergence zone would increase approximately two-fold (fourfold) by 2030 (2060) from the present condition.

[1] Research Institute for Applied Mechanics, Kyushu University, 6-1 Kasuga-Koen, Kasuga 816-8580, Japan. [2] Civil Engineering Research Institute for Cold Region, 1-3-1-34 Toyohira, Sapporo 062-8602, Japan. [3] Tokyo University of Marine Science and Technology, 4-5-7 Konan, Minato-ku, Tokyo 108-8477, Japan. Correspondence and requests for materials should be addressed to A.I. (email: aisobe@riam.kyushu-u.ac.jp)

Recent studies have reported the presence of pelagic plastic fragments with diameters <5 mm (referred to as microplastics) in the open oceans including polar waters[1–6], marginal seas[7,8], and coastal waters[9]. Importantly, microplastics might act as a transport vector of chemical pollutants into the marine ecosystem, due to the absorption of pollutants onto their surfaces[10,11], and their subsequent ingestion by marine organisms as small as zooplankton[12–14]. Furthermore, laboratory-based studies have found that marine organisms are harmed by the ingestion of even additive-free and virgin plastic beads leading to inflammatory responses, impedance of feeding, and so fourth[15–20]. Such damage to marine organisms can be expected in nature given the abundance of mismanaged plastic waste and, hence, pelagic microplastics after fragmentation present in the oceans. The amount of plastic waste in the oceans is expected to increase in the future.

One of the most important aspects of marine plastic pollution research is to quantify microplastic abundance in the present and future ocean environment. Without this, the arguments made by eco-toxicologists and/or environmental chemists may be difficult to justify because of the unrealistic density or sparsity of microplastics in their experimental design. However, mapping the microplastic abundance in the actual ocean has proven difficult, because insufficient measurements are available for all the world's oceans. Numerical model approaches are potentially capable of mapping the pelagic microplastic abundance in the past, present, and future. However, numerical models require careful validation using reliable field measurements and, therefore, the modeling studies published to date have been limited to regions where intensive microplastic surveys have been conducted using a harmonized protocol (e.g., mesh size of sampling nets, wind/wave correction, and unit) over the course of the observations[9,21].

To map the abundance of pelagic microplastics, we conducted a transoceanic survey across a meridional transect from the Southern Ocean[6] to Japan in 2016 (Fig. 1). Stations used in a previous study[3,22] were added to our analyses (Fig. 1 and Supplementary Table 1). Mismanaged plastic waste can escape into the natural environment, especially from countries with a high population density[23]. It is therefore not surprising to find numerous plastic fragments in the oceans of the Northern Hemisphere where plastic debris has degraded on beaches[24]. A synthesis of multiple surveys conducted in the Southern Hemisphere has suggested that pelagic microplastics are less widespread than in the oceans of the Northern Hemisphere[1,2]. Therefore, a remarkable contrast in the abundance of pelagic microplastics is expected across a single meridional transect, while the abundance along zonal transects is likely to be dependent on oceanic conditions, such as convergence zones (i.e., oceanic fronts) that fluctuate in both time and space.

In the present study, the future abundance of pelagic microplastics over the Pacific Ocean was predicted using a numerical model that reproduced the current abundance observed in the meridional survey in 2016. The Pacific Ocean receives the highest amount (52% of the global total; Supplementary Table 2) of mismanaged plastic waste of all the world's oceans. Large amounts of microplastics have been detected around the region described as the Great Pacific Garbage Patch[1,2], and in the East Asian seas[7]. The situation in the Pacific Ocean is likely to occur, eventually in the rest of the world's oceans; thus, predicting marine plastic pollution in this region is a research priority. In addition to the single snapshot campaign in 2016, the decadal variation reported by Goldstein et al.[3] was used to validate the present model by computing the temporal series of oceanic microplastic abundance (Fig. 1 and Supplementary Table 1).

The present study used a numerical tracking model to predict the movements of particles representing non-conservative pelagic microplastics. They were carried by ocean currents, which were provided by an ocean analysis dataset, and by Stokes drift, which was computed using a wave model forced by satellite-derived winds. Pelagic microplastics in the upper ocean are unlikely to act as conservative matter despite their limited decomposition under natural conditions[25]. In addition to transport due to oceanic motion, the ocean plastic circulation is also influenced by plastic removal processes in the upper ocean (hereinafter, referred to as a sink), such as settling to the abyssal ocean after biofouling[26,27], absorption into the marine ecosystem (partly settling along with detritus)[12,28–32], sand beaches[33], and sea ice[34], and fragmentation to small particles[35], which can pass through the sampling nets conventionally used in microplastic surveys (mesh size ~0.3 mm). However, these sinks have not been quantified to date, and it is therefore not easy to establish numerical models that incorporate their effects. To avoid any uncertainty in determining various sinking rates, the model used in this study incorporated a single sink term, by which the modeled particles were removed randomly so that the number ($Q$) of particles released from a source in a year diminished to $Qe^{-t/\tau}$ after time ($t$) from the particles' release. Here $\tau$ is regarded as the average transit time (turnover time)[36] of microplastics in the upper ocean, and is a single adjustable constant chosen to ensure that the modeled particle distribution is consistent with both the meridional survey observations in 2016 and decadal variation recorded by Goldstein et al.[3]. The emission of microplastics should also be incorporated in models because particles are released from sources positioned around the Pacific Ocean. The number of particles released from the various sources was considered to be proportional to the current estimates of mismanaged plastic waste[23], gross domestic product, and future plastic waste in a specific region[23] (Fig. 2; Supplementary Table 2). In this application, modeled particle counts were converted to microplastic abundance in the actual ocean by adjusting the observed abundance at Sta. 38 in Fig. 1 to that modeled at the nearest grid cell (see Numerical modeling in the Methods section for the conversion procedure).

We examined the sources, sinks, and pathways of microplastics in the upper ocean, and computed projected microplastic concentrations for 2066. The results show that most microplastics accumulate in the North Pacific, with the highest concentrations predicted in the East Asian seas and central North Pacific.

## Results

**Abundance of pelagic microplastics along the transect.** The abundance of pelagic microplastics was found to decrease exponentially from the North Pacific, via the equatorial Pacific and Tasman Sea, to the Southern Ocean (Fig. 3). In total, the number of microplastics with diameters < 5 mm (mesoplastics >5 mm) excluding fibers and expanded polystyrene from Sta. 20 to Sta. 38 was 932 (109), accounting for 82% (91%) of those collected across the entire meridional transect. Particle counts per unit seawater volume (hereinafter, concentration) decreased as we moved southward (Fig. 3a). An exponential curve ($C_0 10^{\lambda\varphi}$; $-63.5° < \varphi < 34.2°$ in latitude) was fitted to the concentration in a least square sense, where $C_0$ and $\lambda$ (slope) were 0.02 pieces m$^{-3}$ and 0.0072 degree$^{-1}$, respectively, with a significant correlation coefficient (0.42) suggested by the $t$ test with a 95% confidence level. Particle counts integrated over the water column (hereinafter, total particle count) were less dispersed than the concentrations because of the wind/wave correction (see Methods for the conversion of metrics) (Fig. 3b). An exponential curve fitted to the total particle count of microplastics had $C_0$ and $\lambda$ of $7.6 \times 10^4$ pieces km$^{-2}$ and 0.0097 degree$^{-1}$, respectively, with a correlation coefficient of 0.53, which was significant at the 99% confidence level according to a $t$ test. Averaging the latter exponential curve over the

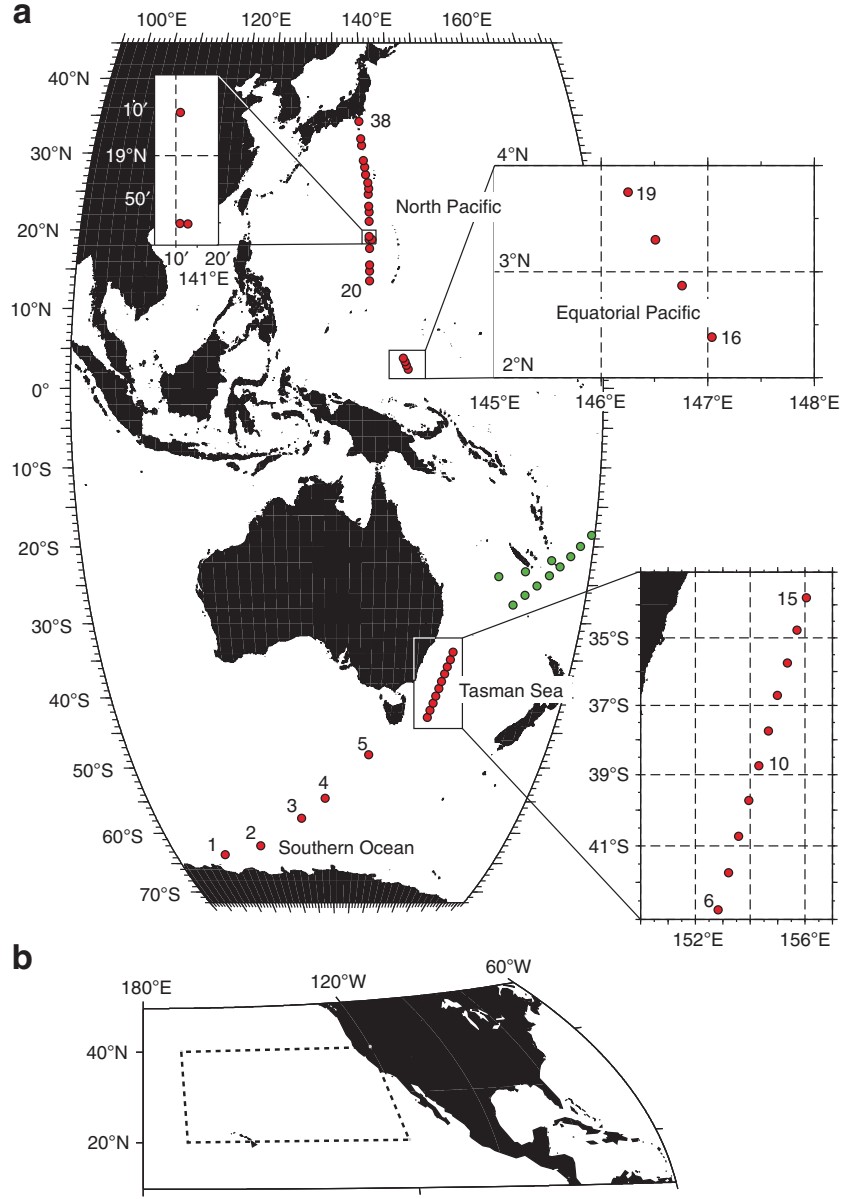

**Fig. 1** Transoceanic microplastic surveys in the present study. The survey stations along a meridional transect from Sta. 1 to Sta. 38 are shown by red dots (**a**). Areas with a dense network of stations are enlarged in the inset maps. The stations used in Reisser et al.[22] are shown by green dots, which complement the large gap between stations 15 and 16. The weight concentrations of microplastics in Goldstein et al.[3] were also used in the present study within the area surrounded by the broken lines (**b**)

latitudes of 30°–40° yielded an approximate total particle count in the mid-latitudes of the South Pacific of around 35,000 pieces km$^{-2}$, an order of magnitude smaller than that in the North Pacific (~160,000 pieces km$^{-2}$). It should be noted that the total particle count at Sta. 38, the northernmost station, reached 8,800,000 pieces km$^{-2}$, which was an order of magnitude larger than in the North Pacific. This was consistent with Isobe et al.[7] who found that the East Asian seas are a hotspot of pelagic microplastics.

**Average transit time of microplastics in the upper ocean.** Overall, the particle-tracking model in the absence of both the sink term (i.e., $\tau \to \infty$) and fishery-based sources did a good job of reproducing the meridional contrast such that the microplastic abundance in the North Pacific was an order of magnitude larger

than that in the South Pacific (Fig. 4); the relatively minor contribution by fishery-based sources are shown later at the end of the Results section. Likewise, in the East Asian seas (>30°N), the modeled abundance that was two orders of magnitude larger than that in the South Pacific is likely to occur in reality, because the abundance averaged over the grid cells north of 30°N (red arrow in Fig. 4) was close to the previous estimate over the East Asian seas around Japan (dot and bar)[7]. The model without the Stokes drift demonstrated that the total particle counts in the North Pacific were shifted northward compared to the model with the Stokes drift (Supplementary Fig. 1). It was therefore suggested that the model underestimated the meridional contrast of the microplastic abundance, especially for less buoyant particles, with smaller sizes and/or a denser polymer composition, moving beneath the layers subject to wind-wave influences.

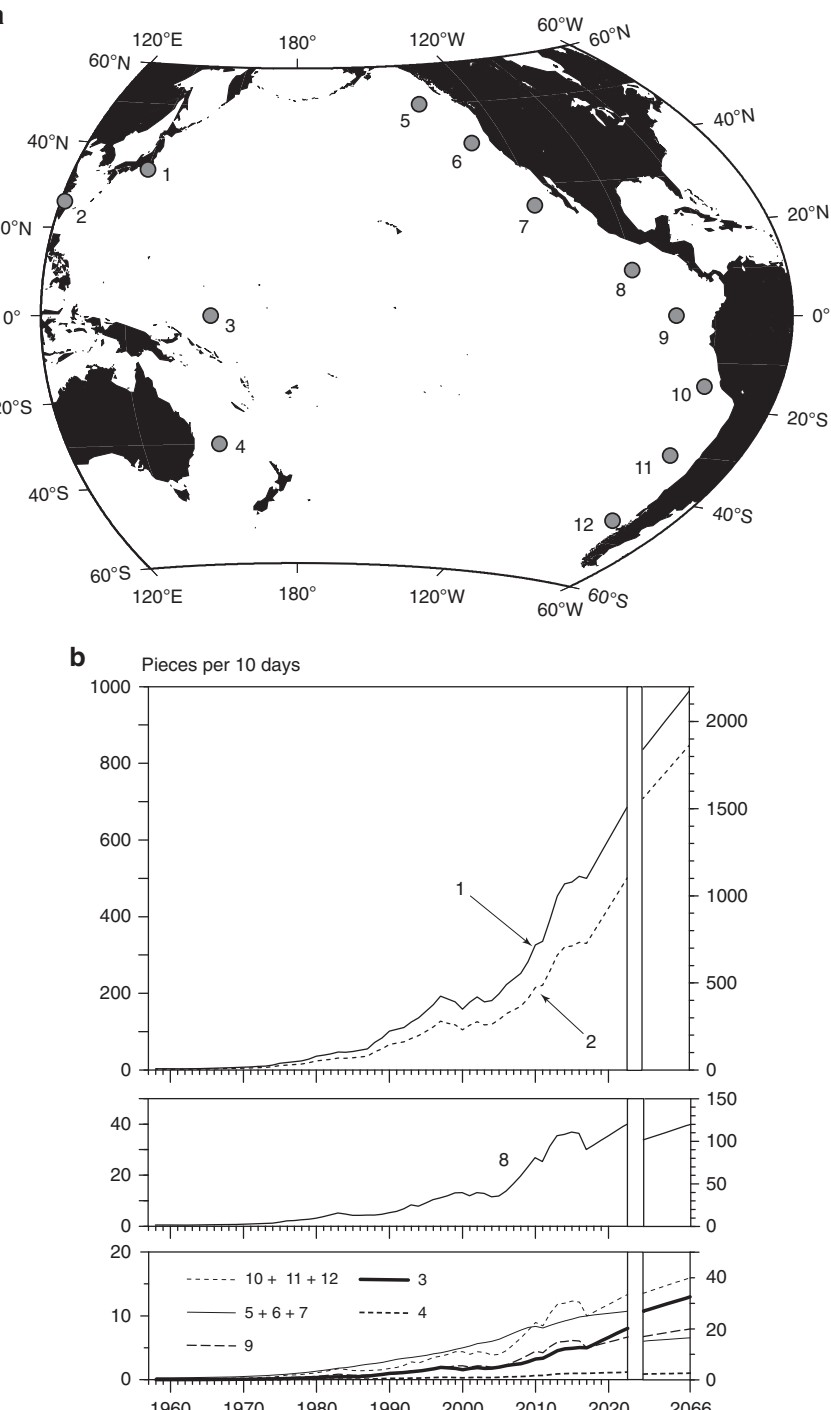

**Fig. 2** Sources of microplastics in the numerical model. Source positions (**a**) and emissions (**b**) are shown for modeled particles representing microplastics during a 110-year computation. The combined emissions in North America (5 + 6 + 7) and South America (10 + 11 + 12) are plotted in **b**

An in-depth comparison between the modeled microplastic abundance and the observations produced a plausible estimate for the average transit time ($\tau$). In the North Pacific where pelagic microplastics were abundant along the transect (Fig. 3), the meridional contrast evaluated by the slope ($\lambda$) of the exponential curve was overestimated as $\tau$ decreased (Fig. 5a). When 1 year was chosen for $\tau$, the slope was 46% larger than the observation, suggesting that microplastic abundance in the model decreased southward more rapidly than projected. The decadal increase of the microplastic abundance measured by the weight per unit seawater volume (weight concentration; see Methods for the conversion of metrics)[3] was overestimated as $\tau$ increased; see Fig. 5b where the abundance increased unrealistically in the model when $\tau$ was longer than 10 years. Therefore, a choice of $\tau$ between 1 and 5 years was likely to be appropriate to minimize the root mean square error of both the slope and decadal variation, when the models with different $\tau$ were compared with observations (Supplementary Table 3). In this application, we chose 3 years for convenience to establish a model that reproduced approximately both the spatial (Fig. 5a) and temporal (Fig. 5b) variations.

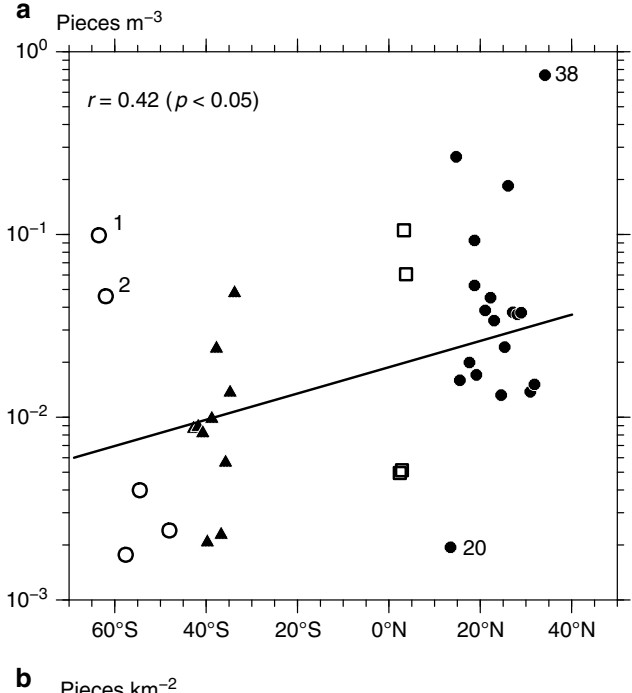

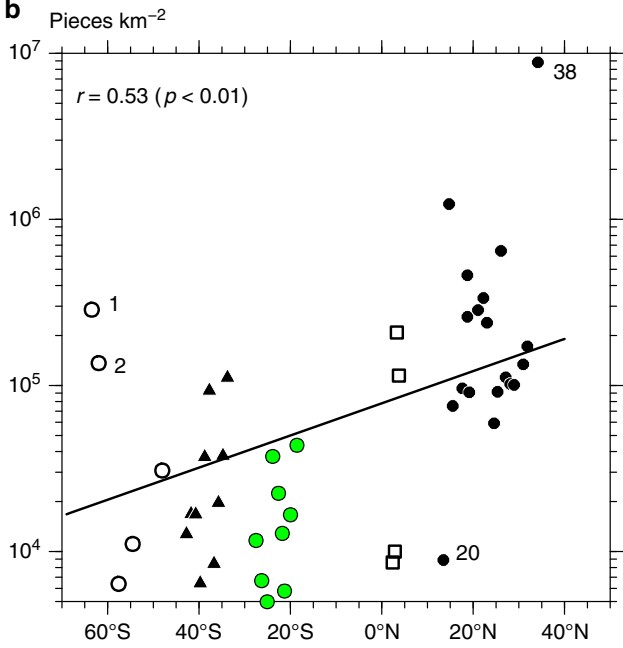

**Fig. 3** Abundance of microplastics along the meridional transect. Panel **a** shows concentrations in the Southern Ocean (open circles), Tasman Sea (closed triangles), Equatorial Pacific (open squares), and North Pacific (closed circles), while **b** is the same but for total particle counts. The digits 1, 2, 20, and 38 by the marks denote the station numbers. The regression lines between locations versus microplastic abundance on a $\log_{10}$-scale are shown in each panel with the correlation coefficient (*r*) and confidence level (e.g., *p* < 0.05 means 95%) suggested by a *t* test. The estimates provided by Reisser et al.[22] are given in **b** (green dots) and show that the different surveys also suggest a remarkable meridional contrast, i.e., an abundance in the South Pacific that was one order of magnitude smaller than that in the North Pacific

**Hindcast/forecast of pelagic microplastics over the Pacific.** Sporadic high weight concentrations were observed over the Pacific Ocean (Fig. 6a, b). The high concentrations occurred mostly at 30°N, with a remarkable seasonality, whereby the weight concentration was higher in the boreal summer due to the

weakened vertical mixing of pelagic microplastics under calm oceanic conditions. In addition, it was found that the weight concentration in the boreal summer was higher in the western part of the study area, along the 30°N band. Weight concentrations in the South Pacific were much smaller than those over the North Pacific, irrespective of the season. This meridional contrast was consistent with the results of a transoceanic survey in 2016 and, thus, both observations and model results indicated that marine plastic pollution was ongoing, particularly in the North Pacific.

From the model prediction, the eccentricity of the pelagic microplastic distribution remained remarkable after 50 years (Fig. 6c, d). In particular, the weight concentrations within three boxes 1, 2, and 3 were higher than that in the surrounding waters in the map for August, 2066. Box 3 is located in the eastern North Pacific where a massive amount of floating macroplastic debris has been reported[37,38]. However, much higher weight concentrations were predicted in boxes 1 (East Asian seas) and 2 (central North Pacific). The weight concentration of surface zooplankton (hence, biological production) was high in boxes 1 and 2 (green stipples in Fig. 6e)[39,40]. Therefore, attention should be given to the western and central Pacific Ocean where marine organisms may encounter environmental risks due to pelagic microplastics in the future.

It is worthwhile to examine how the prediction depends on the model design, because the present study provides part of the process to guide future research on marine plastic pollution. First, the results of the additional experiment using fishery-based sources (Numerical modeling in the Methods section) differed little from the model of only land-based plastic waste around the 30°N-band in the North Pacific (Supplementary Fig. 2). This is because the modeled particle count increase, due to additions from the fishery sources, was ultimately adjusted to the observed abundance at Sta. 38 in the meridional survey in 2016. Nevertheless, in both 2016 and 2066, the weight concentration off South America was slightly higher than that in the model of only land-based plastic waste, because the ratio of fishery-based plastic waste from Chile and Peru accounted for 16.4% of the total amount (Supplementary Table 4), and this ratio was much higher than that of the land-based plastic waste (1.3%; Supplementary Table 2). The contribution of the fishery to pelagic microplastic abundance was not conclusive in the present study due to the ad hoc estimate of plastic waste generated by the fishery. It is nonetheless noteworthy contribution, particularly in waters west of South America. Besides fishery-based plastic wastes, abundance of pelagic microplastics west of South America might be underestimated to some extent because of the lack of microplastic transport from the South Indian Ocean via a surface superconvergence pathway[41] to the subtropical South Pacific gyre. Second, building a 110-year circulation model from only 1 year (2015) of surface circulation might introduce biases in the accumulation pattern of pelagic microplastics. Besides seasonality, interannual to decadal variations such as El Niño and the Pacific Decadal Oscillation are active in the Pacific, and likely to affect surface circulation and, hence, the accumulation pattern[42,43]. However, surface circulation in 2010, when the indices of these variations were in the opposite phase to 2015, also showed intensive convergence around 30°N (subtropical convergence zone) in the western and central (eastern) Pacific in August (February) with nearly the same magnitude (ca. $-3 \times 10^{-6}$ s$^{-1}$; Supplementary Figs. 3 and 4). Thereby, pelagic microplastic accumulation in the western and central (eastern) parts of the subtropical convergence zone in boreal summer (winter) was suggested to be robust against interannual to decadal variation in the Pacific (Supplementary Fig. 5a–d), when our attention was focused on the order of magnitude of microplastic occurrence in

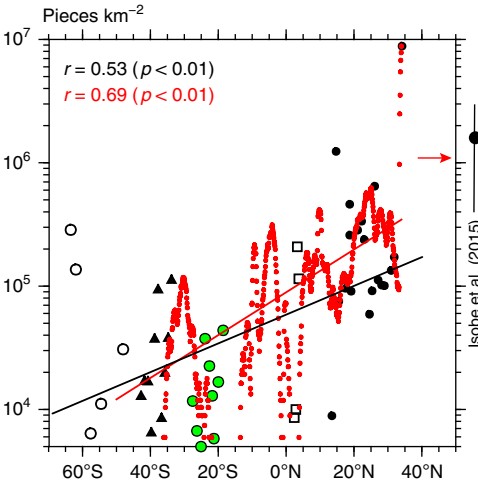

**Fig. 4** Observed and modeled microplastic abundance. The modeled total particle counts are shown by the small red dots along 140 (160)°E in the North (South) Pacific. To compare the modeled abundance with the observed abundance, the modeled total particle count in the grid cell nearest to Sta. 38 was adjusted to that observed at Sta. 38 (see Numerical modeling in the Methods section). The total particle counts in the modeled area north of 30°N were averaged (red arrows) to compare them with the estimate provided by Isobe et al.[7]. The closed circle and bar on the right-hand side of this figure denote the average and standard deviation of total particle counts observed around Japan >30°N, respectively

different broad Pacific regions. Nonetheless, the transition of areas with abundant microplastics (Supplementary Fig. 5e, f) suggested that secular variations in the surface circulation are of critical importance to determine accumulation regions of pelagic microplastics. Third, the delayed emission due to time intervals required for fragmentation from macroplastic debris to microplastics on beaches (see Numerical modeling in the Methods section) might introduce biases in the accumulation pattern of pelagic microplastics. The average transit times in the models with emission delayed by 1, 5, and 10 years were <5 years, which were similar to that in the model without the time intervals in emission (Supplementary Table 5). However, it is found that the average transit time become shorter in the emission delayed by 10 years and, therefore, the choice of 3 years in the present study is not conclusive for life expectancy of pelagic microplastics in the upper layer. Nevertheless, the overall feature of weight concentration maps (Supplementary Fig. 6) was similar to that in Fig. 6.

## Discussion

A lengthy sink term (i.e., persistent microplastics) reproduced well the meridional transect (Fig. 5a), whereas a short term (i.e., non-persistent microplastics) predicted better the long-term evolution in the eastern North Pacific (Fig. 5b). These results suggest that this discrepancy was resolved by setting the life expectancy (3 years in the present model) of microplastics in the upper layer.

We next consider how the non-persistent microplastic scenario performed in the present model. First, although an approximation, the 3-year timescale was nonetheless appropriate to reproduce the long-term evolution determined by the balance between emission and removal of microplastics in the model. Second, the timescale was appropriate to reproduce the meridional transect along which the northern waters contained much more microplastics than those of the south. The abundant microplastics in the north were likely to occur because microplastic emission in the western North Pacific (1 and 2 in Fig. 2) was greatest among

all sources, and because frequent oceanic convergence zones trapped pelagic microplastics around 30°N throughout the year (subtropical convergence zone; Supplementary Fig. 3). Closing the Indonesian Archipelago in the model was unlikely to reduce the amount of microplastics in the south, because the Indonesian Throughflow flowing westward[44,45] prevents microplastics in the Indian Ocean from moving toward the Pacific Ocean. The particle sources and time lapses in the model without the sink term (i.e., $\tau \to \infty$) demonstrated that 60% of the particles along 140°E from the Equator to 15°N in February 2016 originated from Central America (8 in Fig. 2), and that non-negligible fraction of particles had time lapses ranging from 3 to 8 years (Fig. 7). The meridional contrast of microplastics become overestimated when the average transit time chosen for the model was shortened, because microplastics carried westward across the Pacific by the North Equatorial Current were removed more rapidly than in the actual situation.

Averaging the weight concentrations over the boxes 1, 2, and 3 in Fig. 6, we were able to demonstrate the weight concentration of pelagic microplastics in the past, present, and future (Fig. 8). The seasonality of the concentrations was remarkable (Fig. 6), such that the weight concentrations in the North Pacific from April to September exceeded those in the rest of the year. The weight concentrations of pelagic microplastics in the three boxes were predicted to increase approximately twofold (fourfold) by 2030 (2060) from the present condition in 2016. The weight concentration was projected to exceed 1000 mg m$^{-3}$ in summer from the 2060 s onward in parts of the East Asian seas (box 1) and the central North Pacific (box 2) (Figs. 6 and 8a). The model predicted that such high weight concentrations will occur in areas with dense zooplankton, and the concentrations will exceed the weight concentration of suspended particulate matter (SPM) currently reported in the open ocean; 10–600 mg m$^{-3}$ in the Atlantic and two to three times lower in the Pacific[46]. Half of the total SPM (most with diameters < 10 μm) in the North Pacific is non-organic[47] and, therefore, pelagic microplastics are likely to become the predominant non-organic SPM before the 2060s. The weight concentration in the eastern North Pacific (3) increased more slowly than those in the two western boxes (Fig. 8). This was because the sources of microplastics were located mostly in the western North Pacific (Fig. 2), and because the time taken for microplastics to reach the eastern North Pacific was longer than 3 years in the subtropical gyre. Note that the weight concentration in the model without the sink term ($\tau \to \infty$) increased as we moved to the east (blue dots in Fig. 8).

There remains a large gap between microplastic observations (hence, modeling) and laboratory-based studies conducted by eco-toxicologists and/or environmental chemists with respect to both concentrations and sizes of microplastics to which aquatic biota are exposed. Recent laboratory-based studies have demonstrated that plastic beads are harmful to aquatic biota at a variety of concentrations and particle diameters[48] (Fig. 9; Supplementary Table 6 for values and descriptions of each experiment). In depicting Fig. 9 including benthic and/or freshwater organisms as well as pelagic marine organisms, we chose experiments that lacked interaction between microplastics (partly, nanoplastics) and other environmental contaminants, because open oceans such as the Pacific are likely to be less contaminated by pollutants than coastal waters. However, particle sizes were remarkably different from those in the present study, with particles one to four orders of magnitude smaller than those observed and modeled (>0.3 mm). Even if the total mass of microplastics >0.3 mm cloud be conserved after further fragmentation into minute particles, weight concentrations higher than 10$^4$ mg m$^{-3}$ would be unlikely to occur, irrespective of particle sizes in the open ocean within the current century (Fig. 8). The lower limit of

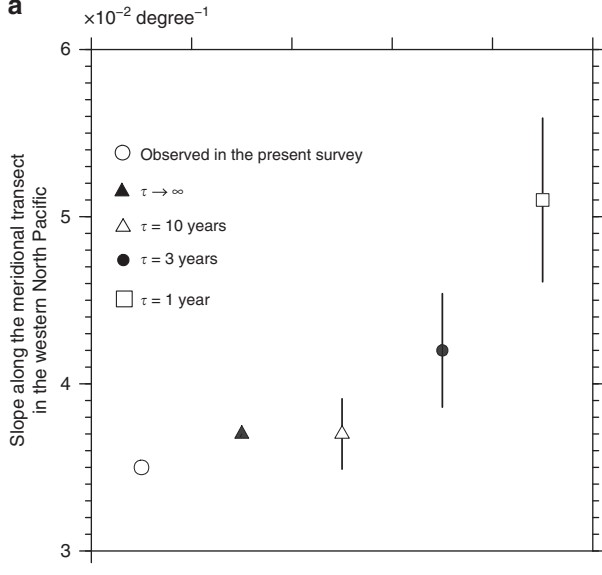

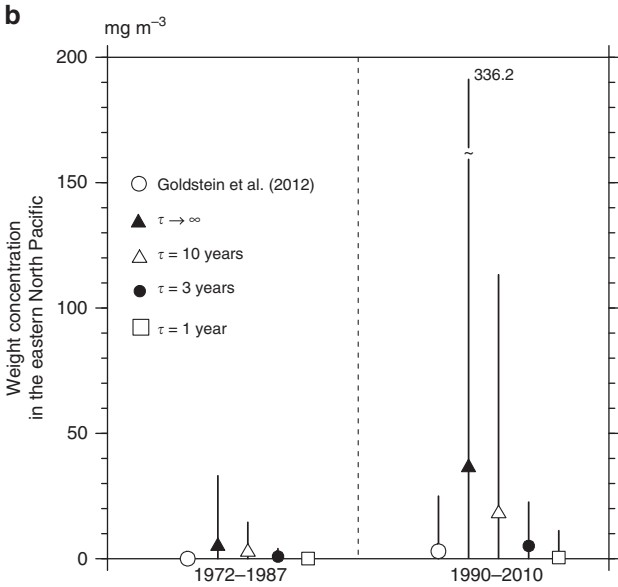

**Fig. 5** Dependence of microplastic abundance on $\tau$. In **a**, the slopes ($\lambda$) of the exponential curve representing the meridional variation of the total particle counts are compared between the meridional transect in 2016 and models with a different average transit time ($\tau$) in the North Pacific. In **b**, weight concentrations averaged between 1972 and 1987 are compared with those averaged between 1990 and 2010 using the estimates given by Goldstein et al.[3] and models with a different $\tau$. The weight concentrations were averaged over the box from 130°W to 170°W, and from 20°N to 40°N. The marks denote the median for each case, while the upper (lower) end of the bars represents the 95th (5th) percentile in line with the estimate by Goldstein et al.[3]. The 95th percentile for the model without the sink (i.e., $\tau \to \infty$) in 1990–2010 is shown by the digit at the top of the bar because the value exceeds the upper limit of **b**

the microplastic concentration harmful to aquatic biota was most frequent (41% in total) between $10^3$ and $10^4$ mg m$^{-3}$ in laboratory-based studies (bar chart in Fig. 9). According to the predictions of the present model, weight concentrations around $10^3$ mg m$^{-3}$ are likely to occur by the 2060s despite the non-persistent properties of microplastics in the upper ocean. Therefore, pelagic microplastics will be potentially harmful to marine organisms exposed to dense concentrations in the western

and central North Pacific, if microplastics as small as those used in the laboratories are generated due to fragmentation, and if the average transit time of these minute microplastics is similar to or longer than that estimated in the present study. However, neither fragmentation to microplastics <0.3 mm nor their removal processes from the upper ocean are conclusive at the present time. Likewise, dense concentrations of microplastics with sizes > 0.3 mm are potentially harmful to fish, due to ingestion, by 2060s, although both effects[49] and uptake[50] are dependent on particle size.

Unless the amount of mismanaged plastic waste is reduced substantially, marine plastic pollution is likely to proceed to a point of no return, beyond which marine organisms will be harmed, as has been shown in laboratory experiments. However, it is necessary to bridge the large gap between the laboratory-based studies and observations/modeling; otherwise the period of the point of no return in nature remains unknown. First, an assimilation of the results from many laboratory-based studies could provide a more precise threshold for the weight concentration of pelagic microplastics, including those with sizes larger than $O(100)$ μm. The results of the present study will be useful for eco-toxicologists and/or environmental chemists when designing future experimental studies. Second, efficient observational techniques capable of quantifying the abundance of microplastics, including those <0.3 mm, should be standardized to monitor the current abundance and to validate the numerical modeling of microplastics in the world's oceans. Third, the numerical model used here, representing different sinks in terms of the average transit time of a single particle, should be improved to create an ocean plastic circulation model, which incorporates several possible sinks, such as settling to the abyssal ocean after biofouling, absorption into the marine ecosystem, sand beaches, and sea ice, as well as fragmentation to minute particles. It is likely that, in reality, these sinks are processes that vary geographically and temporally. Such an ocean plastic circulation model would provide a perspective on marine plastic pollution that could be used in future studies.

## Methods

**Transoceanic survey and analyses**. Surveys in the Southern Ocean (Stas. 1–5 in Fig. 1) were conducted from January 30 to February 4, 2016[6], while surveys at the remaining stations were conducted during the period February 12 to March 2, 2016. SPM including plastic fragments, was collected using a neuston net (5552; RIGO Co., Ltd., Tokyo, Japan) towed by the T/V *Umitaka-maru*, which belongs to the Tokyo University of Marine Science and Technology. Each sample collection took around 40 min, while the ship traveled at a speed of 2–3 knots. The neuston net was positioned about 2 m from the starboard deck of the ship during towing to avoid contamination by plastic fragments from the ship. Floating buoys were attached at the midpoint of the frame of the net mouth; thus, the neuston net was towed immediately below the sea surface (<1-m depth) even in fluctuation due to waves. The mouth dimension, length, and mesh size of the net were 75 × 75 cm, 3 m, and 0.3 mm, respectively. The lower size limit of microplastics considered in this study was determined by this mesh size. A flow meter (5571A; RIGO Co., Ltd.) was installed at the mouth of the net to measure the water volume passing through during sampling. Once the surveys were completed, the flow-meter readings and net mouth dimension were used to calculate the volume of water filtered during each tow. The total seawater volume passing through the net was approximately 19,600 m$^3$ across the entire transect.

The seawater samples including the SPM were sent to Kyushu University for the extraction of plastic fragments. All samples were observed on a monitor display via a USB camera (HDCE-20C; AS ONE Corporation, Osaka, Japan) attached to a stereoscopic microscope (SZX7; Olympus Corporation, Tokyo, Japan) for the visual identification of their color and shape. When the fragments were too small for visual differentiation between microplastics and natural SPM, the polymers in the sampled material were identified using a Fourier transform infrared spectrophotometer (FT-IR alpha; Bruker Optics K.K., Tokyo, Japan). Fibers (probably fishing lines), expanded-polystyrene particles, and natural SPM were removed before any further analyses. Primary microplastics[51], such as pellets, were included in the subsequent analyses despite their small numbers.

The number of plastic pieces remaining were counted in each size category with an increment of 0.1 mm for microplastics, 1 mm for mesoplastics between 5 and 10 mm, and 10 mm for mesoplastics >10 mm. The sizes were defined by the longest

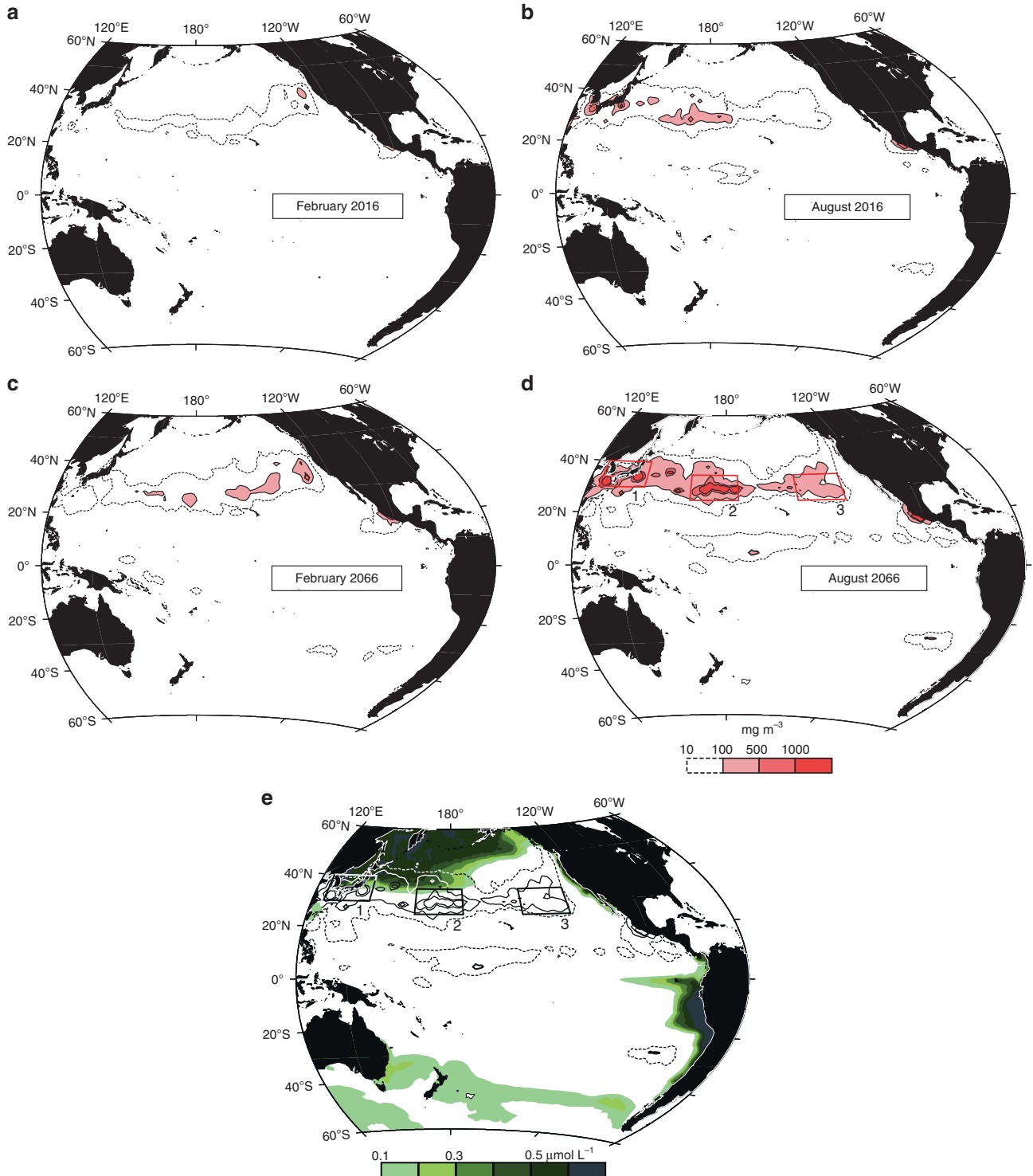

**Fig. 6** Abundance of microplastics in the present and future. The panels represent the weight concentrations averaged in February (**a**) and August (**b**) in 2016, and February (**c**) and August (**d**) in 2066 at the sea surface. The weight concentrations are shown by a red stippling in the line with the scale at the bottom of **d**. The broken curves denote a weight concentration of 10 mg m$^{-3}$. The green stippling in **e** represents surface zooplankton concentrations provided by the Estimated Ocean State for Climate Research dataset[39,40] (see the bottom of the panel for the scale). The zooplankton data were averaged over the boreal spring and summer seasons (April–September) from 2002 to 2011. Also shown by contours in **e** are the weight concentrations in **d**

length of each irregularly shaped fragment visible on the monitor display and were measured using image-processing software (ImageJ downloaded from http:// imagej.nih.gov). Particle counts within each size category were divided by the volume of seawater filtered during each tow to convert them to particle counts per unit seawater volume (concentration with the unit of pieces m$^{-3}$; Supplementary Fig. 7). We obtained the concentration of all microplastics at each station by integrating the concentrations of fragments with sizes ranging from 0.3 to 5 mm.

**Conversion of metrics to a particle count per unit area**. We thereafter converted the microplastic concentration to a total particle count (particle count per unit area; pieces km$^{-2}$) by vertically integrating the concentrations at all depths using the wind speed and significant wave heights measured during each microplastic survey (wind/wave correction). This was undertaken because light-weight micro-plastics are likely to be vulnerable to vertical mixing caused by oceanic turbulence, and because concentrations observed using a neuston net are largely dependent on

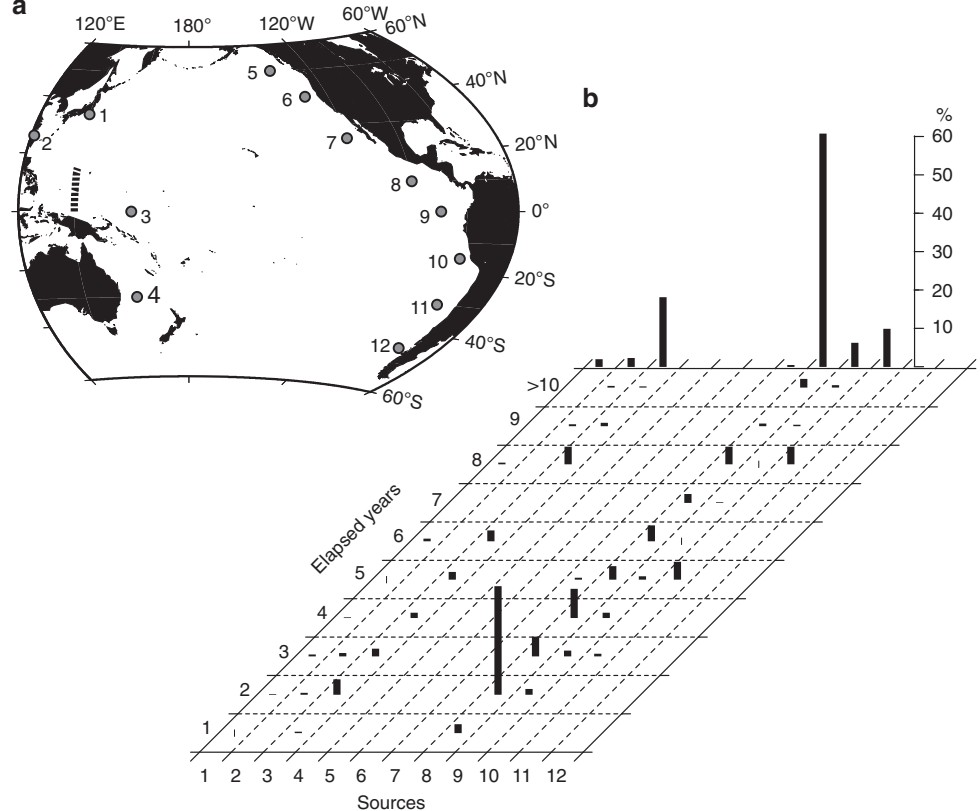

**Fig. 7** Sources and elapsed years of modeled particles. Sources and years elapsed following the release of modeled particles found along 140°E from the Equator to 15°N (bold broken curve in **a**) in February 2016 are shown in **b**. The model without the sink term was used to draw this figure. Source positions are shown in **a**. Bar heights represent percentages of the total particle number across the bold broken curve. Bars at the upper end represent cumulative percentages over all elapsed years at each source

such oceanic conditions[4,52]. The total particle count, regarded as the abundance of pelagic microplastics in the entire water column, is independent of vertical mixing, and it is therefore appropriate to compare microplastic abundance among the different stations along the transect.

The abundance of small plastic fragments decreases exponentially in deeper layers[4,52] and, therefore, the vertical distribution of the microplastic concentration ($N$) can be approximated as follows:

$$N = N_0 e^{\frac{w}{A_0} z}, \qquad (1)$$

where $N_0$ denotes the microplastic concentration at the sea surface ($z = 0$), which was observed using the neuston net; $w$ is the terminal rise velocity of plastics (5.3 mm s$^{-1}$), which was obtained experimentally[4]; and $z$ is the vertical axis measured upward from the sea surface. The parameter $A_0$ is calculated as:

$$A_0 = 1.5 u_* k H_s, \qquad (2)$$

where $u_*$ represents the friction velocity of water ($= 0.0012\, W_{10}$); $k$ is the von Karman coefficient (0.4); $H_s$ is the significant wave height; and $W_{10}$ is the 10-m wind speed[52]. Vertically integrating Eq. (1) from the sea surface ($z = 0$) to an infinitely deep layer ($z \to -\infty$), we can obtain the number of microplastic particles per unit area ($M$) as follows:

$$M = N_0 A_0 / w, \qquad (3)$$

where $M$ is the total particle count mentioned above. Both $H_s$ and $W_{10}$ were recorded once per hour onboard the T/V *Umitaka-maru* over the course of the survey.

In the numerical modeling, we first calculated $M$ and thereafter converted the modeled $M$ to $N_0$ using Eq. (3) with the monthly averaged $H_s$ and $W_{10}$, which were obtained using a wave model and satellite-derived data (see Numerical modeling in this section for details of the procedures and datasets used). In Figs. 5, 6, 8, and 9, the modeled $N_0$ is expressed as the weight per unit seawater volume, with the conversion procedure shown below.

**Conversion of metrics to weight per unit seawater volume.** The modeled particle count in the metrics is expressed as the weight per unit seawater volume in

Figs. 5, 6, 8, and 9 (mg m$^{-3}$). First, the size distribution ($v$) of the microplastics collected in the field survey across the entire meridional transect was approximated to:

$$v(\delta) = \beta \delta e^{-\alpha \delta}, \qquad (4)$$

where $\delta$ denotes the size of the microplastics, and $\alpha$ and $\beta$ were calculated as 0.83 mm$^{-1}$ and $3.8 \times 10^{-2}$ pieces mm$^{-2}$, respectively, in a least square sense (Supplementary Fig. 7; correlation coefficient of 0.89, significant at the 99% confidence level following a $t$ test). Note that $\alpha$ represents the reciprocal of the mode size (1.2 mm). We assumed that an approximation in the form of Eq. (4) was ubiquitously available for the size distribution of microplastics collected using neuston nets, and that the mode size (hence, $\alpha$) was invariant, irrespective of the different situations suggested by previous studies[1,7,9]. Integrating Eq. (4) over the entire microplastic size range ($0.3 < \delta < 5$ mm) provided the surface concentration ($N_0$). When $N_0$ was given by the model, a constant $\beta$ would be uniquely determined by Eq. (4) as follows:

$$\beta = \frac{\int_{\delta_1}^{\delta_2} v\, d\delta}{\int_{\delta_1}^{\delta_2} \delta e^{-\alpha \delta}\, d\delta} = \frac{N_0}{\left[ -\frac{1}{\alpha} \left( \frac{1}{\alpha} + \delta \right) e^{-\alpha \delta} \right]_{\delta_1}^{\delta_2}}, \qquad (5)$$

where $\delta_1$ ($\delta_2$) is the lower (upper) size limit (i.e., 0.3 and 5 mm, respectively), and the operator $[f(\delta)]_{\delta_1}^{\delta_2}$ means $f(\delta_2) - f(\delta_1)$.

We assumed that each microplastic has a cylindrical shape, with a base diameter and height of $\delta$ and $\gamma \delta$, respectively, where $\gamma$ is an adjustable constant (<1.0; shown below) and is a value corresponding to a flat-shaped volume[1]. The weight per unit seawater volume ($W$; weight concentration) was calculated for all microplastics with a size distribution of $v$ as follows:

$$W = \int_{\delta_1}^{\delta_2} \rho \gamma \delta \left( \frac{\delta}{2} \right)^2 \pi v\, d\delta, \qquad (6)$$

where $\rho$ denotes the plastic density (~1.0 g cm$^{-3}$). Substituting Eq. (4) into $v$ in

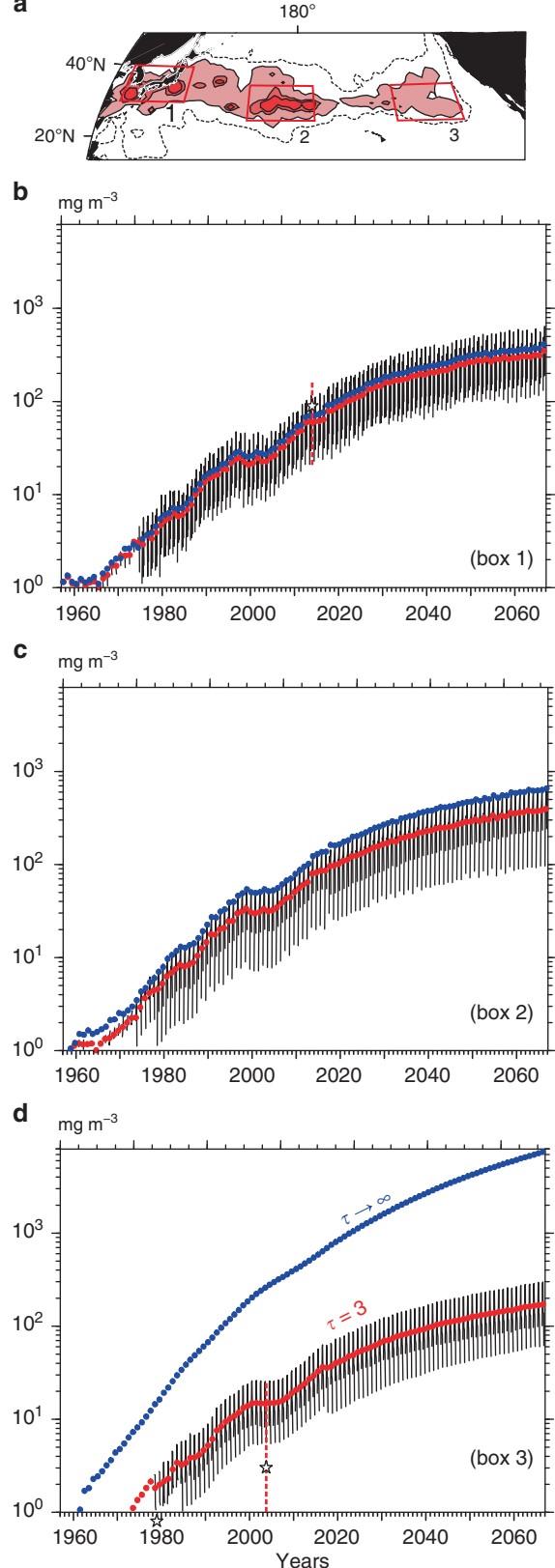

**Fig. 8** One hundred and ten-year variations of the weight concentrations. The weight concentrations of microplastics at the sea surface were averaged over the boxes 1, 2, and 3 in **a**. Also shown in **a** are contours and red stippling of the weight concentrations predicted in August 2066 along the subtropical convergence zone. The monthly averaged concentrations from April to September are shown by the thin curves, while the red dots indicate average for this warming season (**b–d**). Averages for the warming season in the model without the sink term ($\tau \to \infty$) are indicated by the blue dots. The star and broken line in **b** indicate the average and standard deviation of weight concentration observed by Isobe et al.[7], which provided the total particle count averaged over East Asian seas (i.e., box 1) in summer. This total particle count was converted to weight concentration as described in the Methods section. The star and broken line in **d** indicate the median, 5th (lower end) percentile, and 95th (upper end) percentile of weight concentration averaged around box 3 by Goldstein et al.[3]. Weight concentration in 1972–1987 (1999–2010) was plotted at the position of 1979 (2004). Weight concentration in 1972–1987 was smaller than 1.0; therefore, the star was plotted below the abscissa. Weight concentration in 1999–2010 was underestimated in comparison with the model, partly because the concentration provided by Goldstein et al.[3] included data observed in various seasons

or concisely expressed as:

$$W = -\rho\gamma\beta\pi \left[ e^{-\alpha\delta} \sum_{n=1}^{5} \frac{\theta_n \delta^{5-n}}{\alpha^n} \right]_{\delta_1}^{\delta_2}, \qquad (7)$$

where $\theta_n = \theta_{n-1}(6-n)$, $\theta_0 = 0.25$, and $\beta$ is determined by Eq. (5).

The above approximation was considered reasonable when 0.4 was selected for $\gamma$ through trial and error, because the resulting weight concentrations, deduced using Eqs. (6) and (7), were consistent with the those observed directly. The mean weight concentration of microplastics across the entire transect was estimated to be 0.34 mg m$^{-3}$ from the actual weight, measured directly using a mass scale in conjunction with the seawater volume passing through the net at each station. Substituting the surface concentration observed for each size category (bars in Supplementary Fig. 7) into $v$ in Eq. (6), we obtained a weight concentration of 0.30 mg m$^{-3}$ averaged over all stations. This estimate was similar to the observed weight concentration and, thus, the approximation of the shape of a microplastic fragment to a thin cylinder was considered reasonable. Substituting 0.83 mm$^{-1}$ and $3.8 \times 10^{-2}$ pieces mm$^{-2}$, respectively, into $\alpha$ and $\beta$ in Eq. (7), we obtained a weight concentration of 0.29 mg m$^{-3}$, which was close to the observed weight concentration. Therefore, the approximation of Eq. (4) to the size distribution of pelagic microplastics was considered reasonable for calculating the weight concentrations.

**Numerical modeling.** Four models were combined to determine the microplastic distribution over the Pacific Ocean from 60°S to 60°N (Fig. 2), and over the period from 1957 to 2066 (110 years). Our target was microplastics larger than 0.3 mm that we could observe in the present survey, and that we could use to validate model accuracy. The year 1957 was chosen for the beginning of the calculation because the production of plastics has increased substantially over the course of the past 60 years[53], and because the microplastic distribution observed along the transoceanic survey in 2016 was regarded as the current situation. First, an emission model was constructed to generate microplastics from different sources. Second, surface ocean currents were given by oceanic analysis/reanalysis model products. Third, a wave model was used to compute Stokes drift, which also contributes to microplastic displacements in conjunction with surface ocean currents[9,21]. Fourth, a particle-tracking model incorporating a sink term was used to determine the motion of particles representing non-conservative microplastics drifting in the upper ocean.

The emission of microplastics was represented by the particles released from 12 sources surrounding the Pacific Ocean (Fig. 2). The particle counts released in 10-day intervals in the present day (2016) and future (2066) are given in Supplementary Table 2. The positioning of the sources used in this study appears to be over-approximate, and did not adequately capture the distribution of source drivers such as river mouths. The ultimate scope of the present model was, therefore, to reproduce the order of magnitude of microplastic occurrence in different broad Pacific regions. At the time of the study, we had no way of knowing where and how microplastics are generated in the real world and, therefore, we assumed that macroplastic debris dumped in regions (i.e., countries constituting each source; Supplementary Table 2) was fragmentized to microplastics within the same region. In other words, microplastics fragmentized from macroplastic debris moving from remote regions, and/or those fragmentized from macroplastic debris

Eq. (6) enables a conversion from $N_0$ to $W$ as follows:

$$W = -\rho\gamma\beta\pi \left[ e^{-\alpha\delta} \left( \frac{\delta^4}{\alpha} + \frac{4\delta^3}{\alpha^2} + \frac{12\delta^2}{\alpha^3} + \frac{24\delta}{\alpha^4} + \frac{24}{\alpha^5} \right) \right]_{\delta_1}^{\delta_2},$$

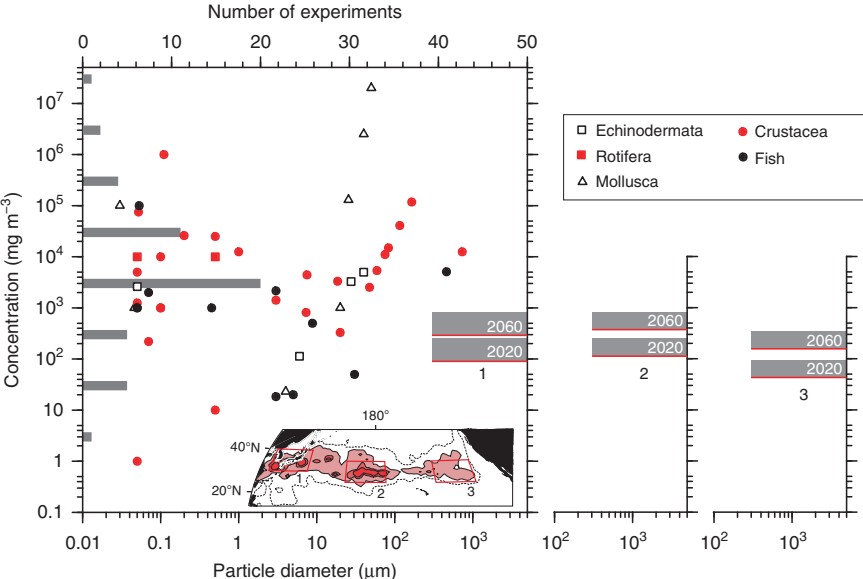

**Fig. 9** Comparison between laboratory-based studies and model. Different marks were used for different classes in the biological classification as shown in the upper right. The 49 experiments listed in Table S1 of de Sá et al.[48] were chosen for this comparison because they reported the weight concentrations of plastic beads in surrounding water to which aquatic biota were exposed. The diameters of plastic beads used for the experiment (lower abscissa) were plotted against the weight concentrations (ordinate). The diameters and minimal weight concentrations that were found to be harmful to organisms in the experiments are plotted, as well as the lower limit of the tested concentrations. When experiments included a range of particle diameters, we used the median value. Gray bars on the left-hand side represent the numbers of experiments by orders of magnitude of the weight concentration (upper abscissa). Red lines indicate modeled weight concentrations averaged over the boxes 1, 2, and 3 in 2020 and 2060; see the inset map for the positions of these boxes and weight concentrations of surface microplastics in August 2066. The height of the gray stippling indicates the average plus one standard deviation. Note that particle diameters in the model ranged from 300 μm (lower limit among observations) to 5 mm (upper limit of the microplastic definition)

drifting in the sea were both assumed to be negligible. The former assumption was justified because the abundance of macroplastic debris littered on beaches increases as the land-based mismanaged plastic waste[23] estimated in the same region increases (Supplementary Fig. 8). The latter assumption was justified because the degradation of macroplastic debris proceeds more rapidly on beaches than in seawater. This is because exposure to ultraviolet radiation and mechanical erosion are minimal when submerged in water[24,54,55], and also because a combination of ocean currents and Stokes drift repeatedly returns the plastic fragments onto beaches until they become microplastics[9]. Therefore, the number of particles currently released was considered to be proportional to the land-based mismanaged plastic waste in each region[23]. The particle counts computed over the model domain were subsequently converted to microplastic abundance using the observed abundance as mentioned at the end of the Numerical modeling section. The number of particles released increased from zero in 1957 to the current estimate in 2016, in proportion with the transition of the gross domestic product summed for each of the 12 regions. Under the assumption that no regulation/operation to reduce the land-based mismanaged plastic waste will be conducted, the expected increase in the number of particles released in the next 50 years was linearly extrapolated using a 15-year prediction of mismanaged plastic waste in each region[23]. The particles were released once in each 10-day period over the course of the computation. The number of modeled particles reached about 4,000,000 by the end of the computation in 2066. The sources assigned for East Asia (1 in Fig. 2) and Southeast Asia (2) were positioned around the Kuroshio Current and Taiwan Warm Current, respectively, because massive amounts of particles are likely to be dispersed due to these ocean currents (hence, surface convergence zones along the oceanic fronts), which intensify along the western boundary of the ocean.

In addition to the land-based plastic waste mentioned above, we examined the contribution of ocean-based plastic waste to microplastic emission. To date, there has been no reliable estimate of ocean-based plastic waste surrounding the Pacific Ocean. Fishery-based plastic debris, which accounted for about 25% (weight), 20% (number), and 10% (number) of macroplastic debris littered on beaches in Japan[56], Australia[57], and the United States[58], respectively, was predominant among ocean-based plastic debris. According to these ratios, the present study set an approximate percentage of 20% between fishery-based and land-based plastic wastes around the Pacific coasts (i.e., 16,467,673 × 10³ kg year⁻¹ [891 particles] × 0.2–3,300,000 × 10³ kg year⁻¹ [180 particles] in Supplementary Table 4). These 180 particles were divided among all sources in proportion to the ratios of fish catches in the neighboring seas of each source to the total catches in all countries around the Pacific coasts (Supplementary Table 4). For simplicity, a time series of fishery-based particle release was determined in the same manner as that of land-based particle release (Fig. 2 and Supplementary Table 2). However, this emission model of fishery-based plastic waste was ad hoc, and

was only a minor contribution to the microplastic abundance in the present model. Therefore, these results are shown separately from those of the model without fishery-based emission, in the Supplementary information.

Furthermore, we examined delayed emission by which time intervals between stranding of macroplastic debris and release of microplastics were given on modeled beaches. This might be justified because macroplastic debris is likely to be fragmentized to microplastics on a timescale of years, although our knowledge on generation of microplastics is very scare. In this additional experiment, numbers of particle release from all sources were proportion with the abundance of macroplastic debris littered in the past; see the particle release delayed by *n* years in Supplementary Fig. 6. This simplification was justified because the abundance of macroplastic debris littered on beaches increases as the land-based mismanaged plastic waste[23] estimated in the same region increases (Supplementary Fig. 8). The computation procedures except the delayed emission were same as those without the delay. The choices of 1, 5, and 10 years for *n* in this delayed emission model were ad hoc, and were a minor contribution to the microplastic abundance in the present model. Therefore, these results are shown separately in the Supplementary information.

Ocean currents and Stokes drift in 2015 were incorporated into the particle-tracking model shown below. The microplastic distribution was observed mostly in February, 2016, and was likely to be reflected by oceanic motion in the latest year. Ocean-current velocities were provided by the Hybrid Coordinate Ocean Model (HYCOM)[59] analysis product for 2015 computation (https://hycom.org/data/glbu0pt08/expt-91pt2), with a resolution of 1/12° in both latitude and longitude; detailed descriptions of the model setup, data assimilation, and forcing are available at the HYCOM website. The University of Miami wave model version 1.0.1[60] was used to compute the Stokes drift over the model domain with 0.25° horizontal resolution. The wave model was driven by wind data acquired by the Advanced Scatterometer, with 0.25° resolution in both latitude and longitude[61]. The ETOPO1[62] provided the bottom topography and coastlines in the wave model. The Stokes drift velocity and significant wave height, both computed in the wave model, were determined once daily during the same period of the HYCOM product. The ocean-current velocities and Stokes drift in 2015, updated once each day, were repeatedly used for the computation over the course of 110 years, and thus, fluctuations of the microplastic distribution due to year-to-year variations in oceanic motion were neglected in the present study. For comparison, the HYCOM reanalysis product for 2010 (https://hycom.org/data/glbu0pt08/expt-19pt1; currently available from 2012) and Stokes drift computed using Advanced Scatterometer wind data for 2010 were used to examine the influence of interannual and decadal variation in surface currents on microplastic distribution. The year 2010 was chosen because the indices for El Niño and Pacific Decadal Oscillation, the most active interannual and decadal variations in the Pacific Ocean, were in the opposite phase to those in 2015 (Supplementary Fig. 4).

To reduce the computational time, a particle tracking model was simplified so that the particles moved on a horizontal two-dimensional plane with a single sink term. The horizontal locations of the modeled particles, $\mathbf{X} = (x, y)$, were updated as follows:

$$\mathbf{X}(t + \Delta t) = \mathbf{X}(t) + \mathbf{U}\Delta t + \frac{1}{2}\left(\mathbf{U} \cdot \nabla_H \mathbf{U} + \frac{\partial \mathbf{U}}{\partial t}\right)\Delta t^2 + R\sqrt{2K_h\Delta t}(\mathbf{i}, \mathbf{j}) + \theta, \quad (8)$$

where $\mathbf{U}$ [$= (u, v)$], $K_h$, $\mathbf{i}$, and $\mathbf{j}$ are the horizontal current vectors, horizontal diffusivity, and unit vectors in the zonal ($x$) and meridional ($y$) directions, respectively[63]. Here, $R$ represents a random number generated at each time step, with an average and standard deviation of 0.0 and 1.0, respectively. The horizontal current velocities were given by a linear combination of the HYCOM ocean currents and Stokes drift at the sea surface ($z = 0$). The microplastics observed in the actual ocean were found in the uppermost layer <1 m[4] and, thus, ocean currents provided by the HYCOM at the surface were considered to be appropriate for reproducing pelagic microplastic motion. However, in general, Stokes drift decreases downward more rapidly than ocean currents and, thus, particle motion induced by the Stokes drift at the surface might deviate from the actual situation, especially for less buoyant microplastics, because of smaller sizes and/or heavier polymer types. Therefore, an alternative modeling without the Stokes drift was also conducted for comparison. Horizontal diffusivity was calculated using the Smagorinsky scheme[64] with the horizontal current velocities. In Eq. (8), $\theta$ is the sink, representing removal processes that operate on pelagic microplastics in the upper ocean. To incorporate the sink in the model, the modeled particles were randomly removed during the course of the computation so that the number ($Q$) of modeled particles released from each source in each year diminished in line with $Qe^{-t/\tau}$, where $t$ is the elapsed time from the particle release and $\tau$ is the adjustable timescale, as mentioned in the text.

The microplastic distribution in the upper ocean was deduced from particle locations updated every 360 s as the time increment ($\Delta t$) in Eq. (8). Modeled particles moving onto land were returned to the ocean at the point where the particles were located one timestep before. In the actual situation, some microplastics washed ashore on beaches would sink into deep layers of sand[33], while others would be returned to the ocean after remaining on the beach for less than a few months[65]. This process of absorption into a beach was included as a sink term ($\theta$) in the model used in this study. Thereafter, the modeled particle count ($P_m$) was converted to the total particle count of microplastics ($M$ in Eq. (3)) over the model domain ($M_m$). We adjusted the total particle count observed at Sta. 38 ($M_{38}$), where microplastic abundance was greatest along the meridional transect in the 2016 survey, to the modeled particle count in the nearest grid cell in February 2016 ($P_{m38}$). Namely, microplastic abundance over the whole domain and period was obtained as follows:

$$M_m = P_m \times \frac{M_{38}}{P_{m38}}. \quad (9)$$

After this conversion, the present model reproduced the observed meridional contrast in microplastic abundance well (Fig. 4). The modeled particle count in a box with a size of 0.1° (1°) in both latitude and longitude was adjusted to depict Figs. 4 and 5 (Fig. 6). The larger boxes removed small-scale disturbances and were used to produce the map of modeled microplastic abundance, shown in Fig. 6.

The above conversion of the modeled particle count to microplastic abundance had the advantage of simplifying the emission model. It is currently very difficult to actually track the emission and formation of secondary microplastics in coastal areas. However, the microplastic abundance reproduced in the present model was independent of the absolute values of particle numbers released from each source (fourth and fifth columns in Supplementary Table 2). Microplastic abundance reproduced in the model would be unchanged even if modeled particle emissions were doubled at all sources, because the particle count doubled over the model domain would ultimately return to that computed in the present model due to the conversion in Eq. (9) (i.e., $M_m = 2P_m \times M_{38}/2P_{m38}$). Of primary importance in the present emission model were ratios of particle numbers released at different sources; we set the minimal particle number at unity at source 4 in 2016 (Supplementary Table 2). These ratios were determined to be consistent under the current estimate of the mismanaged plastic waste[23]. Therefore, the sole adjustable constant in the present model was the average transit time ($\tau$), which adjusted the spatial and decadal gradients of the modeled particle abundance to those observed (Fig. 5 and Supplementary Table 3). The dependence of $\tau$ on weight concentration (Fig. 5) would be invariant even if modeled particle emissions were doubled at all sources, because the spatial and decadal gradients would remain unchanged after the conversion in Eq. (9). Some land-based plastic waste was not included in the current estimate of the mismanaged plastic waste[23]. For example, direct emissions of invisible microplastics from wastewater and catchment run-off were unlikely to be estimated precisely. However, assuming the generation of this unquantified plastic waste on land was in proportion to the present estimate of mismanaged plastic waste[23], as is likely, the relative values of particle numbers released from each source (Supplementary Table 2) will be maintained. Nonetheless, as mentioned above, an additional experiment incorporating fishery-based plastic waste was conducted, because waste associated with fisheries might be out of proportion to land-based mismanaged plastic waste at each source (Supplementary Table 4).

## Data availability

The data used for all figures in the text and Supplementary Information are available from https://doi.org/10.6084/m9.figshare.7532969.v1.

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

## Acknowledgements

The authors sincerely thank the captain, officers, and crew of the *Umitaka-maru* for their assistance during the field surveys. This research was supported by the Environmental Research and Technology Development Fund (4-1502 and SII-2) of the Ministry of the Environment, Japan.

## Author contributions

A.I. directed the project and wrote the manuscript. A.I., K.U., and T.T. conducted the field surveys and analyzed the microplastics. A.I. and S.I. conducted numerical modeling and analyzed the computational results.

## Additional information

**Competing interests:** The authors declare no competing interests.

