## [Peer Review File · Nature Communications]

Reviewers' comments:

Reviewer #1 (Remarks to the Author):

Congratulations on a great manuscript.

This study reports a first full meridional transect of surface microplastics (MPs) concentrations in the Western Pacific Ocean. A numerical model for dispersion of MPs from a weighted source scenario is presented. The modelling framework includes the introduction of a sink term for MPs. A comparison of sink term scenarios is given for two aspects: meridional transect of observations and modelled accumulation in the Eastern Pacific Ocean. Finally, the study presents long term projections and assess the evolution of a threshold value for MPs concentration harming marine ecosystems.

My general recommendation for this manuscript would be to focus the results and discussion section more on the MPs sink term analysis and implication rather than determining a year for ecological threshold exceedance. I trust the observational dataset and the numerical model framework presented here are both innovative contributions to the field. The forecast analysis and risk assessment are somewhat weaker in my opinion. This for two reasons: (1) as you noted the concentration threshold from laboratory-based studies are determined for particles two to three orders of magnitude smaller than the minimum particle size (0.3 mm) from in-situ collections, the sink term for these (nano) particles is unknown and likely different than those of MPs; (2) building a 110-year circulation model from only one year (2015) of surface circulation (re?)-analysis data may introduce biases in accumulation patterns. See for example two publications below that discuss decadal variations in surface water circulation in the North Pacific:

Qiu, B. & Chen, S. Variability of the Kuroshio extension jet, recirculation gyre, and mesoscale eddies on decadal time scales. *J. Phys. Oceanogr.* 35, 2090–2103 (2005).

Howell, E. A., Bograd, S. J., Morishige, C., Seki, M. P. & Polovina, J. J. On North Pacific circulation and associated marine debris concentration. *Mar. Pollut. Bull.* 65, 16–22 (2012).

As such I find some of the conclusions taken from the long-term forecast may not entirely be accurate and it is prejudicial to the manuscript. The discussion on the sink term is very interesting however and I would suggest to further develop the analysis and discussion in this direction. This relates to the question of persistence of MPs at the ocean surface which is still a large unknown in the field. I understand a lengthy term (persistent MPs) reproduces well the meridional transect but a

short sink term (non-persistent MPs) predicts better the long-term evolution in the Eastern Pacific. I would recommend developing on this discrepancy.

Why choosing 3 years for life expectancy? This figure should not be determined from an average of values satisfying the two tests in Figure 5. For example, I understand the short-termed MPs perform bad against your meridional transect because the concentration gradient North to South is overestimated i.e. the North contains much more than the South. Could this be because sources in the South may be underestimated? E.g. perhaps consider including sources around the Indian Ocean which may contribute to your observational area. As such the gradient could be reduced and better match your observations.

There is a conundrum here however as if in that way the non-persistent MPs scenario performs well in both case this may not allow enough time for sources in the Indian Ocean to reach the observational transect (or for sources in the Pacific to reach the Eastern subtropical zone as you noted). This raises the question of sources of MPs: if surface MPs are non-persistent, is the current observed concentration of MPs in the North Pacific a result of recent degradation of accumulating macro-litter at the sea surface. Thus, studying the dispersion of MPs from sources of mismanaged waste to accumulation zones may not be accurate as it should consider the dispersion of macro-litter which likely has a different sink term than MPs. Alternatively, an interpretation of the discrepancy could be justified by geographical variations for sink terms and perhaps model results could be regarded as probabilities to reach certain zones. Regardless, I trust the manuscript would greatly improve if these questions would be further investigated.

Please, also consider below a series of other comments that I trust could help your readers understand your methods better.

With best regards,

Laurent Lebreton

--

Page 11, 1st paragraph: Can you provide the sampling depth? I.e. by how much was the device submerged. Also, can you report the total sampled sea surface area?

Page 15, equation 7: I don't understand the conversion, can you precise what n is for and why it ranges from 1 to 5?

Page 17 Can you please provide more information on which circulation data product you used. Did you run a simulation for the year of 2015 using HYCOM? If so please describe forcing, set-up etc. If the data comes from a third-party provider, the sources should be acknowledged. I suggest adding a description of the model setup and whether it is an analysis or re-analysis product.

Figure 7 This figure should be completed with reported observations from this study and from existing literature, supporting potential scenarios for sink terms.

Reviewer #2 (Remarks to the Author):

Comments to:

"Abundance of non-conservative pelagic microplastics in the upper ocean in the past, present, and future", by: Atsuhiko Isobe et al.

The paper describes an attempt to track the budget of pelagic microplastics in the Pacific Ocean by linking a microplastic emission model and an oceanic transport model. The study attempts to anchor predictions to empirical data by constraining the model to observations of microplastic abundances along a so-called "meridional transect" stretching from Japan to Antarctica.

The study is comprehensive and takes into account, with a generally appropriate, nevertheless coarse, level of detail, the fundamental processes believed to influence the fate and transport of these materials in the oceans.

I found several points that require clarification before this paper could be further considered.

These points are the following:

- 1) The authors deployed a spatially explicit oceanic model; however, due to obvious knowledge gaps, it is currently very difficult to track emissions and formation of secondary microplastics along coastal areas with sufficient detail to make a prediction of spatial distribution. It appears that the authors consider degradation of litter as the only source of oceanic microplastics. However, there are conspicuous direct emissions of microplastics from land too (e.g. from wastewaters and catchment run-off). Most of the particles emitted in this way are smaller than 0.3 mm (considered

here as lower threshold for observed and measured microplastics). The authors have to acknowledge these aspects, explain how different sources are incorporated into their emission model and the significance/implication of overlooking particles smaller than 0.3 mm.

2) Primary emissions of microplastics from land (e.g. Waste water and River runoff) may be responsive to the same drivers used to estimate MP emissions in this study, but the positioning of sources used here appears to be very rough and does not capture adequately the distribution of source drivers (i.e. human population, river estuaries etc). I would appreciate if the Authors could explicit the ultimate scopes of the models (which I believe are to reproduced the order of magnitude of microplastic occurrence in different broad Pacific regions). This will help the readers to follow the study under the correct perspective.

3) The emission and fate model include two largely unknown parameters: microplastic emission/production rate and the sink parameter. The Emission model (based on GDP, population distribution and estimated amounts of mismanaged plastic) is essentially constrained by forcing the estimated number of pelagic particles to be the same as measured in station 38. The concentration in station 38 is however itself the results of advection, diffusion, emissions and sinks. How did the emission and sink parameters could be segregated and constrained individually using only a single forcing equation? In other words, many different combination of emission rate and sink parameter can lead to the same estimated number of particles in Station 38. How did the author decide the best set of these two controlling parameters? The two parameters should be constrained using the full experimental dataset and predictions in all stations, simultaneously. Still it might not yet be granted that accurate estimations of these two parameters can be achieved.

4) As the model is not constrained against a temporal series of oceanic microplastic observation (but only on a single "snapshot campaign") it is crucial to have a very good estimation of the sink parameter to infer on the past and future Oceanic concentrations. As said above the sink parameter is not the only unknown tunable parameter here. Different setting of emission rates, can influence differently the optimization of the sink parameter. It is crucial the authors provide information on their confidence on the parameterization of the sink parameter. This confidence will directly translate into the confidence of hindcasts and forecasts. Therefore this step is essential for assessing the goodness of this study conclusions.

5) The study focuses on modelling microplastic within the 0.3-5 mm range. Model forecasts are used to infer on the future risk with the conclusion that by 2060 marine densities could be so high to affect zooplankton. However, toxicological data cited here for hazard characterization referred to experiments conducted with microbeads 0.02 mm in diameter. It is unlikely that marine zooplankton can effectively or selectively ingest large plastic debris (e.g. higher than 0.3 mm), therefore the type of particles present in the oceans will have a major influence on the ecological implications. Considering these complexity, I think it is not correct to tune-up the conclusion on future risks in the way the authors have done so far.

Replies to Reviewer#1 (Dr. Laurent Lebreton)

Your careful reading and valuable comments are greatly appreciated. Your suggestions are all useful to improve the manuscript. Your comments are written by *italic with the underline*, and our replies follow.

[1] *My general recommendation for this manuscript would be to focus the results and discussion section more on the MPs sink term analysis and implication rather than determining a year for ecological threshold exceedance.*

We agree. We substantially rewrote “Discussion” section so as to focus how the non-persistent microplastic scenario performed in the present model (mainly in the second paragraph, and partly in the third paragraph). In addition, as recommended by the reviewer, we removed determining a year for ecological threshold exceedance (reply [2]). Emphasized in the revised manuscript is not the ecological risk in a specific year as in the former manuscript, but rather a large gap between laboratory-based studies by eco-toxicologists and microplastic surveys/modeling (reply [2], Fig. 9, and the fourth paragraph in “Discussion”). Please see the following replies for detail.

[2] *The forecast analysis and risk assessment are somewhat weaker in my opinion. This for two reasons: (1) as you noted the concentration threshold from laboratory-based studies are determined for particles two to three orders of magnitude smaller than the minimum particle size (0.3 mm) from in-situ collections, the sink term for these (nano) particles is unknown and likely different than those of MPs;*

We agree. We removed a comparison between our model prediction of microplastic abundance and a single eco-toxicological experiment (Cole et al., 2015) in the former manuscript (Fig. 8 in the revised manuscript). Instead, in the revised manuscript, we compared the 49 laboratory-based studies with our model prediction to emphasize a large gap between microplastic observations (hence, modeling) and eco-toxicological experiments with respect to both concentrations and sizes of microplastics. Please see Fig. 9 and descriptions in the fourth paragraph (**P.11, line 13 – P.12, line 15**) in the “Discussion” section.

[3] (2) building a 110-year circulation model from only one year (2015) of surface circulation (re?)-analysis data may introduce biases in accumulation patterns.

We agree with the reviewer suggesting that, interannual to decadal variation such as El Niño and PDO is active in the Pacific, and was likely to affect the surface circulation and, hence, the accumulation pattern. In the revised manuscript, we added the analysis regarding the surface circulation in 2010, when the indices of these variations were in the opposite phase to 2015. It was found that the remarkable convergence was revealed again along the subtropical convergence zone in the western and central (eastern) Pacific in August (February) with nearly the same magnitude as those in 2015, and that the accumulation of pelagic microplastics was suggested to be robust against interannual to decadal variation. Hopefully, interannual and decadal variation in microplastic abundance will be further studied by many researchers stimulated by our modeling. Please see Supplementary Fig. 4 and description as follows:

P.10, line 4 from the bottom – P.11 line 7, “Building a 110-year circulation model from only one year (2015) of surface circulation might introduce biases in...was predicted to be robust against interannual to decadal variation in the Pacific.”

[4] As such I find some of the conclusions taken from the long-term forecast may not entirely be accurate and it is prejudicial to the manuscript.

We appreciate the reviewer for this comment to avoid prejudice to our manuscript. As recommended by the reviewer, we removed determining a year for ecological threshold exceedance in the revised manuscript (reply [2]). In addition to the description on interannual and decadal variations of the surface currents (reply [3]), we added descriptions on limitations of our prediction as follows:

P. 12, line 16 – P. 13, line 10, The last paragraph in the Discussion section.

P.12, the first paragraph, lines 3-5 from the bottom: “However, neither fragmentation to microplastics <0.3 mm nor their removal processes from the upper ocean are conclusive at the

present time.”

P.19, the second paragraph, lines 4-5: “The ultimate scope of the present model was, therefore, to reproduce the order of magnitude of microplastic occurrence in different broad Pacific regions.”

[5] *I understand a lengthy term (persistent MPs) reproduces well the meridional transect but a short sink term (non-persistent MPs) predicts better the long-term evolution in the Eastern Pacific. I would recommend developing on this discrepancy.*

Yes, as the reviewer pointed out, what we should highlight in the present study was this discrepancy. The encouragement and valuable comments by the reviewer are greatly appreciated. As recommended by the reviewer, we added an analysis on how the non-persistent microplastic scenario performed in the present model in the revised manuscript. Please see Fig. 7 newly added in the manuscript, and descriptions in the first and second paragraphs in the Discussion section (**P.9, “Discussion”, line 1- P.10, line 6**).

[6] *Why choosing 3 years for life expectancy? This figure should not be determined from an average of values satisfying the two tests in Figure 5.*

We agree. In the revised manuscript, we added Supplementary Table 3 (estimate of RMS errors) and the description below.

P.7, the second paragraph, lines 1-3 from the bottom: “We chose 3 years to minimize the root mean square error of the slope and decadal variations when the models with different τ (Fig. 5) were compared with observations (Supplementary Table 3).”

[7] *North contains much more than the South. Could this be because sources in the South may be underestimated?*

We appreciate the reviewer for this very interesting comment. In revised manuscript, we carefully re-examined the origin(s) of modeled particles, and added the discussion why “the North

contains much more than the South”. Please see the second paragraph in the “Discussion” section in the revised manuscript (**P.9, lines 12 from the bottom – P.10, line 6**), and Fig. 7 newly added to the manuscript. We found that the modeled particles (hence, pelagic microplastics) in the South were mostly carried westward across the Pacific by the North Equatorial Current in the model without the sink term, and the short-termed sink term (< 1 year) removed microplastics more rapidly than in the actual situation.

[8] *E.g. perhaps consider including sources around the Indian Ocean which may contribute to your observational area.*

We added the below sentence in the revised manuscript.

P. 9, lines 1-4 from the bottom: “Closing the Indonesian Archipelago in the model was unlikely to reduce the amount of microplastics in the south, because the Indonesian Throughflow flowing westward^{41,42} prevents microplastics in the Indian Ocean from moving toward the Pacific Ocean.”

[9] *There is a conundrum here however as if in that way the non-persistent MPs scenario performs well in both case this may not allow enough time for sources in the Indian Ocean to reach the observational transect (or for sources in the Pacific to reach the Eastern subtropical zone as you noted).*

As suggested by the reviewer, of particular importance in this study was that non-persistent microplastic scenario does not allow enough time for some sources to reach other areas. For instance, the microplastics from Central America cannot reach the south of the western North Pacific (second paragraph in “Discussion” [**P.9, lines 12 from the bottom – P.10, line 6**], and Fig. 7). The microplastics from Asia cannot reach area *c*, eastern North Pacific as mentioned below.

P. 11, lines 1 – 6 from the bottom of the first paragraph: “The weight concentration in the eastern North Pacific (*c*) increased more slowly than those in the two western boxes (Fig. 8). This was because the sources of microplastics were located mostly in the western North Pacific (Fig. 2),

and because the time taken for microplastics to reach the eastern North Pacific was longer than 3 years in the subtropical gyre. Note that the weight concentration in the model without the sink term ($\tau \rightarrow \infty$) increased as we moved to the east (blue dots in Fig. 8).”

[10] if surface MPs are non-persistent, is the current observed concentration of MPs in the North Pacific a result of recent degradation of accumulating macro-litter at the sea surface. Thus, studying the dispersion of MPs from sources of mismanaged waste to accumulation zones may not be accurate as it should consider the dispersion of macro-litter which likely has a different sink term than MPs.

(P. 19, lines 6 – 7 of the second paragraph) “At the time of the study, we had no way of knowing where and how microplastics are generated in the real world” and, therefore, assumptions for the generation of microplastics were required to conduct numerical modeling on microplastics. First of all, we would be grateful if the reviewer would understand the current situation.

Nonetheless, in the revised manuscript, we rewrote our two assumptions more carefully by adding new figure (Supplementary Fig. 6).

In the present study, **(P.19, lines 13 - 15)** “microplastics fragmentized from macroplastic debris moving from remote regions, and/or those fragmentized from macroplastic debris drifting in the sea were both assumed to be negligible.”

Regarding the former assumption, we added Supplementary Fig. 6 to demonstrate that abundance of macroplastic debris littered on beaches increases as the land-based mismanaged plastic waste (Jambeck et al. 2015²³) estimated in the same region increases. Our numerical model approach would be criticized if macro-litter carried remotely was included in the microplastic generation despite the locality of litter in Supplementary Fig. 6. The below description was also added in the revised manuscript.

P. 19, lines 7-9 from the bottom: “The former assumption was justified because the abundance of

macroplastic debris littered on beaches increases as the land-based mismanaged plastic waste²³ estimated in the same region increases (Supplementary Fig. 6).”

Regarding the latter assumption, we described as:

P. 19, lines 2 – 7 from the bottom: “The latter assumption was justified because the degradation of macroplastic debris proceeds more rapidly on beaches than in seawater. This is because exposure to ultraviolet radiation and mechanical erosion are minimal when submerged in water^{24, 53,54}, and also because a combination of ocean currents and Stokes drift repeatedly returns the plastic fragments onto beaches until they become microplastics⁹.”

[11] *an interpretation of the discrepancy could be justified by geographical variations for sink terms and perhaps model results could be regarded as probabilities to reach certain zones.*

Although we also consider that the incorporation of geographical variation for sink terms is likely to improve the ocean plastic circulation model, this is apparently beyond the technical limitation at the present time. We would appreciate it if the reviewer would understand the situation that such modeling framework remains as a next target. We added the below sentence in the revised manuscript.

P. 13, lines 1 – 3 from the bottom of “Discussion”: “It is likely that, in reality, these sinks are processes that vary geographically and temporally. Such an OPCM would provide a perspective on marine plastic pollution that could be used in future studies.”

[12] *Page 11, 1st paragraph: Can you provide the sampling depth? I.e. by how much was the device submerged. Also, can you report the total sampled sea surface area?*

We added the below descriptions. We consider that the seawater volume passing through the net (according to flowmeter read) was more useful than total sampled sea surface area because this excludes influences of ocean currents, but the area is nonetheless readily computed by using the net

dimension given in the manuscript.

P. 13, lines 1 – 3 from the bottom: “Floating buoys were attached at the midpoint of the frame of the net mouth; thus, the neuston net was towed immediately below the sea surface (< 1-m depth) even in fluctuation due to waves.”

P.14, lines 5 - 6: “The total seawater volume passing through the net was approximately 19,600 m³ across the entire transect.”

[13] Page 15, equation 7: I don’t understand the conversion, can you precise what n is for and why it ranges from 1 to 5?

We rewrote as follows. The equation (7) in the former manuscript included a mistake, and so we revised it. The careful reading by the reviewer is greatly appreciated.

P. 17, lines 1- 6 from the bottom: “Substituting Eq. (4) into v in the Eq. (6) enables a conversion from N_0 to W as follows:

$$W = -\rho\gamma\beta\pi \left[e^{-\alpha\delta} \left(\frac{\delta^4}{\alpha} + \frac{4\delta^3}{\alpha^2} + \frac{12\delta^2}{\alpha^3} + \frac{24\delta}{\alpha^4} + \frac{24}{\alpha^5} \right) \right]_{\delta_1}^{\delta_2},$$

or concisely expressed as:

$$W = -\rho\gamma\beta\pi \left[e^{-\alpha\delta} \sum_{n=1}^5 \frac{\theta_n \delta^{5-n}}{\alpha^n} \right]_{\delta_1}^{\delta_2}, \quad (7)$$

where $\theta_n = \theta_{n-1}(6 - n)$, $\theta_0 = 0.25$, and β is determined by Eq. (5). ”

[14] Page 17 Can you please provide more information on which circulation data product you used. Did you run a simulation for the year of 2015 using HYCOM? If so please describe forcing, set-up etc. If the data comes from a third-party provider, the sources should be acknowledged. I suggest adding a description of the model setup and whether it is an analysis or re-analysis product.

The HYCOM product in 2015 is an analysis dataset. We added the below sentence in the revised manuscript.

P. 21, the second paragraph, lines 3 - 7: “Ocean-current velocities were provided by the Hybrid Coordinate Ocean Model (HYCOM)⁵⁸ analysis product for 2015 computation (<https://hycom.org/data/glbu0pt08/expt-91pt2>), with a resolution of 1/12° in both latitude and longitude; detailed descriptions of the model setup, data assimilation, and forcing are available at the HYCOM website.”

In addition, we added the below sentence for 2010 data added to the revised manuscript for comparison.

P. 21, line 1 from the bottom – P. 22, line 3: “For comparison, the HYCOM reanalysis product for 2010 (<https://hycom.org/data/glbu0pt08/expt-19pt1>; currently available from 2012) and Stokes drift computed using ASCAT wind data for 2010 were used to examine the influence of interannual and decadal variation in surface currents on microplastic distribution.”

[15] *Figure 7 This figure should be completed with reported observations from this study and from existing literature, supporting potential scenarios for sink terms.*

We agree. Plots from existing literature (Isobe et al., 2015⁷; Goldstein et al., 2012³) were added in Fig. 8 (former Fig. 7).

Replies to Reviewer#2

Your constructive comments are greatly appreciated. Your suggestions are all useful to improve the manuscript. Your comments are written by *italic with the underline*, and our replies follow.

[1] *The authors deployed a spatially explicit Oceanic model however, due to obvious knowledge gaps, it is currently very difficult to track emissions and formation of secondary microplastics along coastal areas with sufficient detail to make prediction of spatial distribution. It appears that the authors consider degradation of litter as the only source of oceanic microplastics. However, there are conspicuous direct emissions of microplastics from land too (e.g. from wastewaters and catchment run-off). Most of particles emitted in this way are smaller than the 0.3 mm (considered here as lower threshold for observed and measured microplastics). The authors have to acknowledge these aspects, explain how different sources are incorporated into their emission model and the significance/implication of overlooking particles smaller than 0.3 mm.*

We would appreciate it if the reviewer notes that our target was microplastics larger than 0.3 mm that we could observe, and that we did not explicitly discuss minute microplastics smaller than 0.3 mm in the manuscript. To avoid misleading the readers, we added the below sentence in the revised manuscript.

P. 18, “Numerical modeling”, lines 3 - 4: “Our target was microplastics larger than 0.3 mm that we could observe in the present survey, and that we could use to validate model accuracy.”

However, regardless of particle sizes, we recognized the importance of this comment that we have to address particularly in two contexts. In the revised manuscript, (a) we added descriptions to avoid misleading the readers, and (b) we added an extra experiment incorporating microplastic sources that were not included in the former manuscript.

(a) First, we would be grateful if the reviewer would understand that the modeled emissions from the sources included “direct emissions” (e.g., from wastewater and catchment run-off) as well as the emissions by degradation of macro-litter on beaches, because the modeled particle count was

converted to microplastic abundance by using the actually-observed abundance of microplastics, regardless of where/how they generated (rather, it was impossible to specify). However, we found that descriptions regarding both emission and conversion were quite insufficient to convey our model design to the readers.

A new paragraph was therefore added in the **(P. 24, line 1 – P. 25, line 2)** “Method, numerical modeling” in the revised manuscript to explain how we converted the modeled particle count to microplastic abundance in the ocean, and how we incorporated emissions of microplastics including land-based tiny fragments from wastewater and catchment run-off. The essence of the new paragraph is as follows.

In the present study, the particle count computed in the model (P_m) was converted to the microplastic abundance (M_m : “total particle count” in this manuscript) over the whole domain and period as follows:

$$M_m = P_m \times \frac{M_{38}}{P_{m38}}, \quad (9) \text{ (newly added to the revised manuscript)}$$

where M_{38} is the total particle count of microplastics observed at Sta. 38, and P_{m38} is modeled particle count in the nearest grid cell to Sta. 38 in February 2016. Please note that microplastic abundance reproduced in the present model was independent of the absolute values of particle numbers released from each source. The microplastic abundance reproduced in the model (M_m) would be unchanged even if modeled particles emissions were doubled at all sources, because the particle count doubled over the model domain would ultimately return to that computed in the present model due to the conversion in Eq. (9) (i.e., $M_m = 2P_m \times M_{38}/2P_{m38}$). Of primary importance in the present emission model were not absolute values, but ratios between particle numbers released at different sources. These ratios were determined to be consistent under the current estimate of the land-based mismanaged plastic waste (Jambeck et al., 2015) and, therefore, the ratios were not tunable. As suggested by the reviewer, direct emissions of invisible microplastics from wastewater and catchment run-off were unlikely to be estimated precisely. However, assuming the generation of this unquantified plastic waste on land was in proportion to the present estimate of mismanaged plastic waste²³, as is likely, the relative values of particle numbers released from each source (Supplementary Table 1) will be maintained.

(b) Second, as suggested by the reviewer, we nonetheless recognized that the generation of microplastics that had not been included in our former manuscript should be carefully re-considered.

In the revised manuscript, an additional experiment incorporating ocean-based plastic waste derived mainly from fishery was conducted, because waste associated with fisheries might be out of proportion to land-based mismanaged plastic waste at each source (Supplementary Table 2). The emission from the fishery-based waste was described in the third paragraph (**P. 20, line 9 from the bottom – P. 21, line 7**) added newly in the “Method, numerical modeling” section. The results were described in the last paragraph of “Results, Hindcast/forecast of pelagic microplastics over the Pacific” (**P. 8, line 8 from the bottom – P. 9, line 4**) and Supplementary Fig. 2 both newly added in the revised manuscript

[2] Primary emissions of microplastics from land (e.g. Waste water and River runoff) may be responsive to the same drivers used to estimate MP emissions in this study, but the positioning of sources used here appears to be very rough and does not capture adequately the distribution of source drivers (i.e. human population, river estuaries etc). I would appreciate if the Authors could explicit the ultimate scopes of the models (which I believe are to reproduced the order of magnitude of microplastic occurrence in different broad Pacific regions). This will help the readers to follow the study under the correct perspective.

We agree. The sentences recommended by the reviewer were added as follows.

P. 19, lines 6 - 9: “The positioning of the sources used in this study appears to be over-approximate, and did not adequately capture the distribution of source drivers such as river mouths. The ultimate scope of the present model was, therefore, to reproduce the order of magnitude of microplastic occurrence in different broad Pacific regions.”

[3] The emission and fate model include two largely unknown parameters: microplastic emission/production rate and the sink parameter. The Emission model (based on GDP, population

distribution and estimated amounts of mismanaged plastic) is essentially constrained by forcing the estimated number of pelagic particles to be the same as measured in station 38. The concentration in station 38 is however itself the results of advection, diffusion, emissions and sinks. How did the emission and sink parameters could be segregated and constrained individually using only a single forcing equation? In other words, many different combinations of emission rate and sink parameter can lead to the same estimated number of particles in Station 38. How did the author decide the best set of these two controlling parameters? The two parameters should be constrained using the full experimental dataset and predictions in all stations, simultaneously. Still it might not yet be granted that accurate estimations of these two parameters can be achieved.

We added descriptions to avoid misleading the readers.

This is related to our reply to comment 1) regarding the emission. We would grateful if the reviewer understands that the emission in the present model was NOT adjusted by the observation at Sta. 38, because it was technically impossible to “force the estimated number of pelagic particles to be the same as measured in Sta. 38” (it was 8,800,000 particles/km² much larger than the total number of the particles released in the model). We just converted the particle count in the model domain to microplastic abundance by Eq. (9) in Reply (1a). Please understand our model design that the emission and sink parameters were therefore segregated and constrained individually, and please note again that modeled results would not change even if the emission at all sources were doubled due to the conversion by Eq. (9) as shown above. In the revised manuscript, we emphasized that the average transit time (τ) was the unique tunable constant in the present model as follows.

P. 24, lines 9 - 17: “Of primary importance in the present emission model were ratios of particle numbers released at different sources; we set the minimal particle number at unity at source 4 in 2016 (Supplementary Table 1). These ratios were determined to be consistent under the current estimate of the mismanaged plastic waste²³. Therefore, the sole adjustable constant in the present model was the average transit time (τ), which adjusted the spatial and decadal gradients of the modeled particle abundance to those observed (Fig. 5 and Supplementary Table 3). The dependence of τ on weight concentration (Fig. 5) would be invariant even if modeled particle emissions were

doubled at all sources, because the spatial and decadal gradients would remain unchanged after the conversion in Eq. (9).”

[4] As the model is not constrained against a temporal series of oceanic microplastic observation (but only on a single “snapshot campaign”) it is crucial to have a very good estimation of the sink parameter to infer on the past and future Oceanic concentrations. As said above the sink parameter is not the only unknown tunable parameter here. Different setting of emission rates, can influence differently the optimization of the sink parameter. It is crucial the authors provide information on their confidence on the parameterization of the sink parameter. This confidence will directly translate into the confidence of hindcasts and forecasts. Therefore, this step is essential for assessing the goodness of this study conclusions.

We added descriptions to avoid misleading the readers.

Please note Fig. 5 and (P.6) “Results, Average transit time of microplastics in the upper ocean” section. The present model was constrained against decadal gradient (Goldstein et al., 2012³) of oceanic microplastic observation, not only on a single snapshot campaign. Rather, highlighted in the present study was that (P. 9, “Discussion”, lines 1 – 3) a lengthy sink term (i.e., persistent microplastics) reproduced well the meridional transect (panel a), whereas a short term (i.e., non-persistent microplastics) predicted better the long-term evolution in the eastern North Pacific (panel b). To avoid misleading the readers, we added the below descriptions in the revised manuscript. In addition, as recommended by the Reviewer#1, we estimated the RMS errors to show the confidence on the parameterization of the sink parameter.

P. 4, line 3 from the bottom – P. 5, line 1: “Here, τ is regarded as the average transit time (turnover time)³⁶ of microplastics in the upper ocean, and is a single adjustable constant chosen to ensure that the modeled particle distribution is consistent with both the meridional survey observations in 2016 and decadal variation recorded by Goldstein et al. (2012)³.”

P. 9, “Discussion”, lines 1 – 3: “Figure 5 shows that a lengthy sink term (i.e., persistent microplastics) reproduced well the meridional transect (panel a), whereas a short term (i.e.,

non-persistent microplastics) predicted better the long-term evolution in the eastern North Pacific (panel *b*).”

P. 7, lines 1 – 3 from the bottom of the second paragraph: “We chose 3 years to minimize the root mean square error of the slope and decadal variations when the models with different τ (Fig. 5) were compared with observations (Supplementary Table 3).”

[5] The study focuses on modelling microplastic within the 0.3-5 mm range. Model forecasts are used to infer on the future risk with the conclusion that by 2060 marine densities could be so high to affect zooplankton. However, toxicological data cited here for hazard characterization referred to experiments conducted with microbeads 0.02 mm in diameter. It is unlikely that marine zooplankton can effectively or selectively ingest large plastic debris (e.g. higher than 0.3 mm), therefore the type of particles present in the oceans will have a major influence on the ecological implications. Considering these complexity, I think it is not correct to tune-up the conclusion on future risks in the way the authors have done so far.

We agree. We removed a comparison between our model prediction of microplastic abundance and a single eco-toxicological experiment (Cole et al., 2015) in the former manuscript (Fig. 8 in the revised manuscript). Instead, in the revised manuscript, we compared the 49 laboratory-based studies with our model prediction to emphasize a large gap between microplastic observations (hence, modeling) and eco-toxicological experiments with respect to both concentrations and sizes of microplastics. Please see Fig. 9 and descriptions in the fourth paragraph (**P.11, line 13 – P.12, line 15**) in the “Discussion” section.

Reviewers' comments:

Reviewer #1 (Remarks to the Author):

I am more satisfied with this version of the manuscript as the authors decided to focus the discussion on microplastic sink term and implications of their results rather than determining a specific year for reaching an ecological threshold. As such the authors addressed my main concern and I trust the manuscript greatly benefit from this consideration. However, I still have two major comments that I think were not sufficiently addressed in this new version despite acknowledgement in the discussion:

(1) I find the numerical modelling approach still not convincing. I strongly suggest using multiple years for sea surface current and wave data while modelling the transport of buoyant plastic particles. I appreciate the effort in using another year (2010) to compare results. However, these results need to be presented. The equivalent of Figure 6 but using data from 2010 should be depicted at least in supplementary material. A sensitivity analysis and a difference plot between accumulations using data from 2010 and 2015 should also be proposed. Ideally, the authors should be using the full available reanalysis data from HYCOM (20+ years) when producing prediction over 110 years.

(2) The justification for a sink term with turnaround period of 3 years is not conclusive. The minimization of RMS for the sum of latitudinal and temporal gradients is not a good approach in my opinion. Basically, the authors show contradicting results and I find the formulation of an 'in between' solution from persistent and non-persistent microplastics not convincing. For instance, the authors do not consider the time required for the formation of microplastics and assume the direct production of such particles from the generation of mismanaged waste. However, while the authors rightfully discuss that the formation of microplastics is likely coming from the degradation of stranded macroplastics on shores, no consideration of time interval between stranding of plastic waste and the representative release of microparticles is given. This process could take years if not decades to be representative of accumulated waste. Maybe considering this time lag could solve the discrepancy with the decadal evolution in the Eastern Pacific as presented by Goldstein et al. Also the justification for the contribution of Indian Ocean sources into the South Pacific is not correct in my opinion, debris from the Indian Ocean can reach the South Pacific through the Great Australian bight then the Tasman sea and not through the Indonesian Throughflow as justified by the authors. See Maes et al 2018, A Surface "Superconvergence" Pathway Connecting the South Indian Ocean to the Subtropical South Pacific Gyre.

Reviewer #2 (Remarks to the Author):

The manuscript has improved considerably.

Minor changes are still required as the presentation is not clear in some crucial points. Please see the attached comments to the reviewers reply for details.

Replies to Reviewer#1 (Dr. Laurent Lebreton)

Your careful reading is again greatly appreciated. We addressed all of your concerns as follows. Your comments are written by *italic with the underline*, and our replies follow.

[1] I strongly suggest using multiple years for sea surface current and wave data while modelling the transport of buoyant plastic particles. I appreciate the effort in using another year (2010) to compare results. However, these results need to be presented. The equivalent of Figure 6 but using data from 2010 should be depicted at least in supplementary material. A sensitivity analysis and a difference plot between accumulations using data from 2010 and 2015 should also be proposed. Ideally, the authors should be using the full available reanalysis data from HYCOM (20+ years) when producing prediction over 110 years.

We agree. As recommended by the reviewer, we added the equivalent of Figure 6 but using data from 2010 in supplementary material (Supplementary Fig. 5). In addition, a difference plot between accumulations using data from 2010 and 2015 was depicted (right panels in Supplementary Fig. 5). We added the below descriptions related to this revision.

P. 9 line 2 from the bottom – P.10 line 6: “Thereby, pelagic microplastic accumulation in the western and central (eastern) parts of the subtropical convergence zone in boreal summer (winter) was suggested to be robust against interannual to decadal variation in the Pacific (Supplementary Fig.5), when our attention was focused on the order of magnitude of microplastic occurrence in different broad Pacific regions. Nonetheless, the transition of areas with abundant microplastics (right panels in Supplementary Fig. 5) suggested that secular variations in the surface circulation are of critical importance to determine accumulation regions of pelagic microplastics.”

We appreciate it if the reviewer understands that our attention was not focused on specifying accumulation regions (which may depend on secular variations of ocean currents), but on the order of magnitude of microplastic occurrence in different broad Pacific regions as mentioned above (underline).

[2] The justification for a sink term with turnaround period of 3 years is not conclusive. The minimization of RMS for the sum of latitudinal and temporal gradients is not a good approach in my opinion. Basically, the authors show contradicting results and I find the formulation of an 'in between' solution from persistent and non-persistent microplastics not convincing.

We agree. The choice of 3 years might sound conclusive in the former manuscript. We reworded as follows.

P. 7, lines 5-1 from the bottom: “Therefore, a choice of τ between 1 and 5 years was likely to be appropriate to minimize the root mean square error of both the slope and decadal variation, when the models with different τ were compared with observations (Supplementary Table 4). In this application, we chose 3 years for convenience to establish a model that reproduced approximately both the spatial (Fig. 5a) and temporal (Fig. 5b) variations.”

P. 10, lines 3-4 from the bottom of “Results” section: “the choice of 3 years in the present study is not conclusive for life expectancy of pelagic microplastics in the upper layer.”

[3] For instance, the authors do not consider the time required for the formation of microplastics and assume the direct production of such particles from the generation of mismanaged waste. However, while the authors rightfully discuss that the formation of microplastics is likely coming from the degradation of stranded macroplastics on shores, no consideration of time interval between stranding of plastic waste and the representative release of microparticles is given. This process could take years if not decades to be representative of accumulated waste. Maybe considering this time lag could solve the discrepancy with the decadal evolution in the Eastern Pacific as presented by Goldstein et al.

As suggested by the reviewer, we conducted additional experiments including the time interval between stranding of macroplastic debris and release of microplastics (Supplementary Table 5 and Supplementary Fig. 6). The related descriptions were added as follows.

P. 22, second paragraph: “Furthermore, we examined “delayed emission” by which time intervals between stranding of macroplastic debris and release of microplastics were given on modeled beaches. This might be justified because macroplastic debris is likely to be fragmented to microplastics on a timescale of years, although our knowledge on generation of microplastics is very scarce. In this additional experiment, numbers of particle release from all sources were proportion with the abundance of macroplastic debris littered in the past; see the particle release delayed by n years in Supplementary Fig.6. This simplification was justified because the abundance of macroplastic debris littered on beaches increases as the land-based mismanaged plastic waste²³ estimated in the same region increases (Supplementary Fig. 8). The computation procedures except the delayed emission were same as those without the delay. The choices of 1, 5, and 10 years for n in this “delayed emission model” were *ad hoc*, and were a minor contribution to the microplastic abundance in the present model. Therefore, these results are shown separately in the Supplementary information.”

P. 10, lines 6-13: “Third, the delayed emission (see “Numerical modeling” in the Methods section) might introduce biases in the accumulation pattern of pelagic microplastics. The average transit time in the models with emission delayed by 1, 5, and 10 years was less than 5 years, which were similar to that in the model without the time intervals in emission (Supplementary Table 5). However, it is found that the average transit time become shorter in the emission delayed by 10 years and, therefore, the choice of 3 years in the present study is not conclusive for life expectancy of pelagic microplastics in the upper layer. Nevertheless, the overall feature of weight concentration maps (Supplementary Fig. 6) was similar to that in Fig. 6.”

[4] Also the justification for the contribution of Indian Ocean sources into the South Pacific is not correct in my opinion, debris from the Indian Ocean can reach the South Pacific though the Great Australian bight then the Tasman sea and not through the Indonesian Throughflow as justified by the authors.

First, please note that the slopes in Fig. 5a were computed using microplastic abundance only in the North Pacific where the concentrations were sufficiently high for computing statistics. Thus, the superconvergence pathway that conveys materials from the South Indian Ocean to the South Pacific was very difficult to be referred along with

the Indonesian through flow. In the revised manuscript, we emphasized the computation only in the North Pacific as follows:

P. 7, second paragraph, lines 2-3: “In the North Pacific where pelagic microplastics were abundant along the transect (Fig. 3), the meridional contrast evaluated by the slope (λ) of the exponential curve was overestimated as τ decreased (Fig. 5a).”

Fig. 5. Dependence of microplastic abundance on the average transit time (e -folding time, τ , in the text) in the upper ocean. In the upper panel, the slopes (λ) of the exponential curve representing the meridional variation of the total particle counts are compared between the meridional transect in 2016 and models with a different τ in the North Pacific.

Second, we however recognized that a surface transoceanic pathway should be referred when we mention the microplastic abundance in the South Pacific. The revision was as follows.

P. 9, lines 11-14: “Besides fishery-based plastic wastes, abundance of pelagic microplastics west of South America might be underestimated to some extent because of the lack of microplastic transport from the South Indian Ocean via a surface “superconvergence” pathway⁴⁵ to the subtropical South Pacific gyre. ”

Replies to Reviewer#2

Your careful reading is again greatly appreciated. We addressed your concerns as follows. Your comments are written by *italic with the underline*, and our replies follow.

1) The text initially focus a lot on the campaign and give particular emphasis to Reiser data. The same emphasis is not given to Goldstein data. Please correct.

To avoid misleading the readers, we added Supplementary Table 1 in the revised manuscript. The periods and regions of microplastic surveys in the present study, Reisser et al. (2013), and Goldstein et al. (2012) were all included in this table. The related descriptions added to the revised manuscript are as follows.

P. 3, second paragraph, lines 2-3: “Stations used in a previous study^{3,22} were added to our analyses (Fig. 1 and Supplementary Table 1)”

2) Figure 1 should include all the data used to calibrate the model. Therefore Goldstein data position should be displayed, as done for Reisser data.

We added Goldstein data positions in Figure 1.

Fig. 1. Microplastic surveys used in the present study. The survey stations along a meridional transect from Sta.1 to Sta. 38 are shown by the red dots. Areas with a dense network of stations are enlarged in the inset maps. The stations used in Reisser et al. (2013)²² are shown by green dots which complement the large gap between stations 15 and 16. Data used by Goldstein et al. (2013)³ were also used in the present study within the area surrounded by the broken lines in the upper panel.

3) Figure 3 and 4 should show make a distinction between data in the North Pacific coming from this study and taken from Goldstein

Please note that Goldstein data were obtained in the eastern North Pacific, while data in the western North Pacific (along our survey line) were plotted in Figures 3 and 4. To avoid misleading the readers, we added Supplementary Table 1 to clearly show the

observation regions.

4) Figure 5 panel b should also show a comparison with data from the present study and from Reisser et al. (I believe)

Please note that Reisser data cannot be plotted in Figure 5b because their campaigns were conducted in 2011-2012. To avoid misleading the readers, we added Supplementary Table 1 to clearly show the observation periods.

In addition Figure 5 is not entirely clear. Vertical axes should be labelled in the charts. Where in the lower panel are the comparisons with data from the 2012-2016 campaigns? Why only comparison with Goldstein data are reported?

We added titles on vertical axes in Figure 5. Figure 5b was depicted using Goldstein data along with the modeled results. Please note that there were no field campaigns from 2012-2016 in this manuscript. To avoid misleading the readers, we added Supplementary Table 1 to clearly show the observation periods.

REVIEWERS' COMMENTS:

Reviewer #1

Thank you for adressing all my comments.

Congratulations on a much improved manuscript.